# Ecological and social pressures interfere with homeostatic sleep regulation in the wild

J Carter Loftus[1,2,3,4,5,6]\*, Roi Harel[2,5], Chase L Núñez[2,4,5], Margaret C Crofoot[1,2,3,4,5,6]\*

[1]Department of Anthropology, University of California, Davis, Davis, United States; [2]Department for the Ecology of Animal Societies, Max Planck Institute of Animal Behavior, Konstanz, Germany; [3]Department of Biology, University of Konstanz, Konstanz, Germany; [4]Centre for the Advanced Study of Collective Behaviour, University of Konstanz, Konstanz, Germany; [5]Mpala Research Centre, Nanyuki, Kenya; [6]Animal Behavior Graduate Group, University of California, Davis, Davis, United States

\*For correspondence:
jcloftus@ucdavis.edu (JCL);
mcrofoot@ab.mpg.de (MCC)

Competing interest: The authors declare that no competing interests exist.

**Abstract** Sleep is fundamental to the health and fitness of all animals. The physiological importance of sleep is underscored by the central role of homeostasis in determining sleep investment – following periods of sleep deprivation, individuals experience longer and more intense sleep bouts. Yet, most sleep research has been conducted in highly controlled settings, removed from evolutionarily relevant contexts that may hinder the maintenance of sleep homeostasis. Using triaxial accelerometry and GPS to track the sleep patterns of a group of wild baboons (*Papio anubis*), we found that ecological and social pressures indeed interfere with homeostatic sleep regulation. Baboons sacrificed time spent sleeping when in less familiar locations and when sleeping in proximity to more group-mates, regardless of how long they had slept the prior night or how much they had physically exerted themselves the preceding day. Further, they did not appear to compensate for lost sleep via more intense sleep bouts. We found that the collective dynamics characteristic of social animal groups persist into the sleep period, as baboons exhibited synchronized patterns of waking throughout the night, particularly with nearby group-mates. Thus, for animals whose fitness depends critically on avoiding predation and developing social relationships, maintaining sleep homeostasis may be only secondary to remaining vigilant when sleeping in risky habitats and interacting with group-mates during the night. Our results highlight the importance of studying sleep in ecologically relevant contexts, where the adaptive function of sleep patterns directly reflects the complex trade-offs that have guided its evolution.

## Editor's evaluation

By recording sleep and movements in a group of baboons, this study reveals ecological and social drivers of sleep patterns in the wild. Using accelerometry, rather than methods that map brain activity, enables the investigation of animal sleep under natural conditions across a wide range of taxa.

## Introduction

Sleep is an important and understudied facet of animal lives, with every species, from honey bees to humans, allocating a portion of each day to this period of rest (*Cirelli and Tononi, 2008*). The universality of sleep reflects its central role in important physiological processes, including memory

consolidation, support of the central nervous system, energy conservation, and physical restoration (*Chowdhury and Shafer, 2020*; *Gangwisch, 2014*; *Stickgold, 2005*; *Vyazovskiy, 2015*). Accordingly, failure to meet daily sleep demand has health consequences (*Basner et al., 2013*), with potentially fatal repercussions of extreme sleep deprivation (*Rechtschaffen and Bergmann, 2002*). The physiological need for sleep is emphasized by its homeostatic control – after periods of insufficient sleep or extreme physical exertion, individuals experience particularly long and intense bouts of sleep (*Kitamura et al., 2016*). Decades of sleep research have consistently implicated homeostasis as a primary determinant of sleep patterns, such that homeostatic regulation has become an important criterion in the very definition of sleep (*Siegel, 2008*).

A strong focus on studying sleep in the laboratory or at the bedside, although revealing much about the physiology of sleep, has inherently overlooked the ecological pressures that drive the regulation and evolution of sleep (*Rattenborg et al., 2017*; *Reinhardt, 2020*). In the natural world, the significance of sleep extends beyond its direct physiological impacts. Sleeping animals typically cannot engage in other behaviors that are important to their survival (but see *Lyamin et al., 2008*; *Lyamin et al., 2018*; *Rattenborg et al., 1999*; *Rattenborg et al., 2016*), and investing in sleep when environmental forces render vigilance and activity particularly important may impose substantial costs to wild animals. In addition to preventing animals from foraging, searching for mating opportunities, defending territories, and caring for young, sleep leaves animals in a state of extreme inattention, and thus highly vulnerable to their predators (*Lima et al., 2005*). The evolution of sleep and its manifestation in the wild may therefore be driven by a complex balance between the physiological need for sleep and ecological costs imposed on sleeping animals.

For gregarious animals, the balance between the costs and benefits of sleep may be further modulated by the social environment. However, even the most basic aspects of sleeping with conspecifics, such as whether the social context facilitates or constrains sleep, remain unknown (*Karamihalev et al., 2019*). Sleeping in a social context could alter the costs of sleep – the sentinel hypothesis suggests that staggering the timing of sleep bouts in a group can collectively maintain both high-quality sleep and high levels of antipredator vigilance (*Samson et al., 2017*; *Snyder, 1966*). Sleeping in a group may therefore facilitate an individual's ability to fulfill its physiological sleep requirement by reducing the risk of doing so. Alternatively, social dynamics may actually inhibit investment in sleep. Sleep may present social opportunity costs, causing individuals to sacrifice sleep in order to spend more time actively engaging with group-mates. Additionally, proximity to group-mates may cause sleep disruptions initiated by short periods of wakeful activity of neighboring individuals. Cascading disruptions could then lead to collective dynamics of sleep, such as the waves of wakefulness that have been documented in flocks of gulls (*Beauchamp, 2009*; *Beauchamp, 2011*). Thus, sleeping in close proximity to conspecifics may potentially be accompanied by both costs and benefits for an individual's ability to obtain sufficient sleep, and discovering how these potential costs and benefits are actually realized will shed light on the forces that have guided sleep adaptations in social animals.

To understand how group-living animals navigate trade-offs between their physiological need for sleep and the ecological and social pressures that shape the costs and benefits associated with this biological imperative, we investigated the factors shaping sleep patterns of wild olive baboons (*Papio anubis*). Baboons live in stable multi-male, multi-female groups of up to 100 individuals (*Cheney and Seyfarth, 2008*), and during the night, they seek safety in trees and rock outcroppings (*Altmann and Altmann, 1970*; *Busse, 1980*). Despite seeking refuge in these sleep sites, baboons remain particularly vulnerable to nighttime predation from leopards, which represents the single largest source of mortality for adult baboons (*Cheney et al., 2004*; *Cowlishaw, 1994*; *Isbell et al., 2018*). Baboons must therefore navigate the trade-off between investing in sleep and maintaining antipredator vigilance. As a highly gregarious animal whose fitness depends heavily on their social relationships (*Silk et al., 2009*), baboons must also balance their time spent sleeping with their investment in social interactions as time constraints during the day limit their ability to build and maintain their relationships (*Dunbar, 1992*).

We simultaneously tracked the activity of 26 wild olive baboons from the same group using collars fitted with GPS sensors and triaxial accelerometers to understand how baboons manage their competing nighttime priorities. Accelerometer-based sleep classification has shown an impressive ability to detect and monitor sleep behavior across taxa (*Ancoli-Israel et al., 2003*; *de Souza et al., 2003*; *Hoffmann et al., 2012*; *Ladha and Hoffman, 2018*; *Malungo et al., 2021*; *Qin et al., 2020*),

and is now commonly used to assess sleep in both humans (e.g., *Jones et al., 2019*; *Patel et al., 2017*) and nonhuman animals (e.g., *Gravett et al., 2017*; *Reinhardt, 2020*; *Samson et al., 2018*). Validation studies comparing performance of this noninvasive method to polysomnography – the gold standard in sleep research – generally show high accuracy (78–90%; *Ancoli-Israel et al., 2003*; *Kanady et al., 2011*; *Malungo et al., 2021*; *Shambroom et al., 2012*), although concerns remain about the ability of movement-based methods to distinguish sleep from resting wakefulness (*Ancoli-Israel et al., 2003*; *de Souza et al., 2003*), and results must be evaluated with these caveats in mind. For this study, we adapted a well-validated sleep classification algorithm used in human research (*van Hees et al., 2015*; *van Hees et al., 2018*) and validated its ability to detect sleep in wild baboons. We then used this algorithm to describe the sleep patterns of members of our study group over a period of a month (*Supplementary file 1a*). We used these data to assess the influence of homeostatic regulation on the pattern of sleep and wake bouts within nights, as well as the duration and fragmentation of sleep across nights. We also leveraged naturally occurring sleep disturbances to test how recent sleep history influenced arousal threshold during sleep. We compared the influence of homeostatic regulation to that of the location in which individuals slept (both within the sleep site as well as between distinct sleep sites) and their local social environment, both of which may exert pressures on sleep behavior in the wild that conflict with the maintenance of homeostasis.

## Results

The diel pattern of activity in wild baboons, as reflected by accelerometry data, reveals a clear monophasic sleep pattern, with individuals active during the day and inactive at night (*Figure 1B*). To derive metrics of sleep (sleep onset time, awakening time, total sleep time, sleep period duration, sleep efficiency, sleep fragmentation), we calculated the log of the vectorial dynamic body acceleration (VeDBA), a widely used measure of overall movement activity (*Qasem et al., 2012*), from 36 calendar days or 354 baboon-nights. Sleep onset occurred 53.0 ± 1.7 (mean ± SE) min prior to the end of evening astronomical twilight, and baboons awoke 35.9 ± 1.7 min after the beginning of morning astronomical twilight (*Figure 1A and C*). The duration of the sleep period – the period between sleep onset and awakening – was 11.0 ± 0.04 hr on average. Within the sleep period, baboons slept for an average of 9.2 ± 0.04 hr (total sleep time), displaying an average sleep efficiency of 85.0% ± 0.2%. Baboons exhibited 1.8 ± 0.03 distinct wake bouts per hour of sleep during the sleep period (sleep fragmentation).

Due to high correlation of total sleep time with onset time, awakening time, sleep period duration, and sleep efficiency, as well as a strong relationship between sleep fragmentation and sleep efficiency (*Supplementary file 1b*), we focused the majority of our analyses on total sleep time and sleep fragmentation. Individuals differed in their total sleep time and sleep fragmentation, and much of this variation reflected differences between the sexes and variation across age categories. Males slept an average of 20 min longer than females and experienced 0.23 fewer wake bouts per hour of sleep (total sleep time linear mixed model [LMM]: standardized estimate [95% credible interval lower bound, 95% credible interval upper bound]: 0.44 [–0.05, 0.94]; sleep fragmentation LMM: –0.44 [–1.04, 0.17]). Juveniles and subadults slept, on average, 16 min less and woke on 0.38 and 0.28 more occasions per hour of sleep, respectively, than adults (total sleep time LMM: juveniles: –0.34 [–1.15, 0.50], subadults: –0.34 [–0.81, 0.14]; sleep fragmentation LMM: juveniles: 0.72 [–0.28, 1.75], subadults: 0.52 [–0.06, 1.10]).

An individual's recent history of sleep and activity was not a strong driver of sleep patterns (*Figure 2*, *Figure 2—figure supplement 1*, *Figure 2—figure supplement 2*). Neither total sleep time nor sleep fragmentation the previous night affected time spent napping the following day, time spent sleeping the following night, or the fragmentation of sleep the following night (*Supplementary file 1f*; time spent napping LMM: previous night total sleep time: 0.03 [–0.10, 0.17], previous night sleep fragmentation: 0.00 [–0.11, 0.12]; *Figure 2A* i-ii, total sleep time LMM: previous night relative total sleep time: –0.04 [–0.19, 0.12], previous night relative sleep fragmentation: –0.06 [–0.19, 0.06]; *Figure 2B* i-ii, sleep fragmentation LMM: previous night relative total sleep time: 0.09 [–0.10, 0.27], previous night relative sleep fragmentation: –0.03 [–0.18, 0.12]). However, after spending more time napping during the day, baboons did experience shorter total sleep time and more fragmented sleep during the night (*Figure 2Ai and Bi*; total sleep time LMM: –0.19 [–0.36, –0.01], sleep fragmentation LMM: 0.22 [0.00, 0.45]). For every 10 min spent napping, baboons spent six fewer minutes sleeping

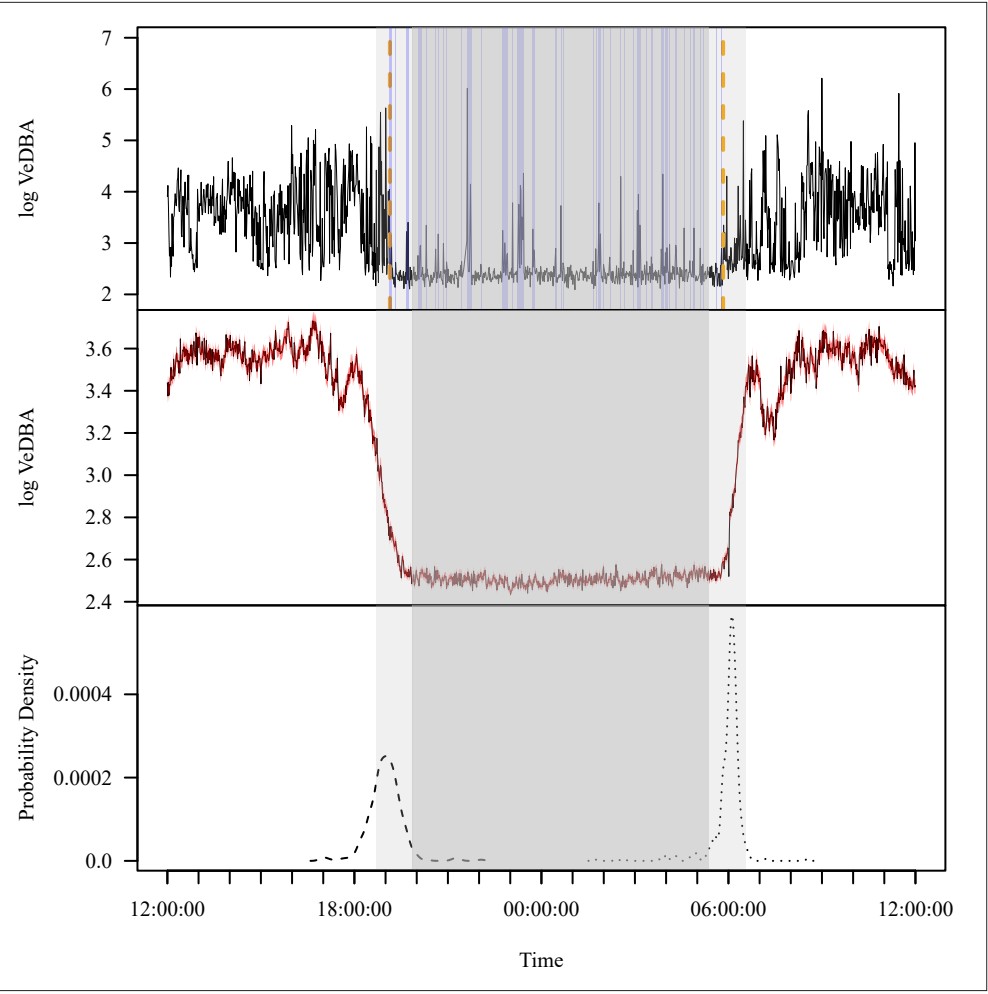

**Figure 1.** Extracting activity and sleep from accelerometry in a group of wild olive baboons. Adapting algorithms developed by *van Hees et al., 2015* and *van Hees et al., 2018*, we used the vectorial dynamic body acceleration (VeDBA), a measure of overall activity, to determine the sleep onset and awakening times (**A**; orange dashed lines), as well as periods of wake after sleep onset (**A**; blue shading) for each individual baboon on each day. These metrics allowed us to calculate the total sleep time, sleep period duration, sleep efficiency, and sleep fragmentation. The plot (**A**) shows the data of one individual within a single noon-to-noon period as an example. Averaged across all individuals on all nights (N = 354 baboon-nights), the log VeDBA shows that baboons exhibit activity patterns typical of a diurnal animal with monophasic sleep (**B**), with a consolidated period of very low levels of activity during the night. Although the timing of waking (**C**; dotted line) was more consistent across the group and across the study period than the timing of sleep onset (**C**; dashed line), both sleep onset and waking typically occurred within astronomical twilight. The red shading in (**B**) indicates ±1 SE. In all subplots, the gray shaded region depicts the period between sunset and sunrise, with double shading from the end of evening astronomical twilight to the beginning of morning astronomical twilight.

the following night and woke 0.10 times more per hour of sleep. Neither sleep duration nor fragmentation was influenced by the distance baboons traveled (*Figure 2Aiv and Biv*; total sleep time LMM: –0.03 [–0.19, 0.14]; sleep fragmentation LMM: –0.04 [–0.13, 0.05]) or the VeDBA they accumulated during the day (*Supplementary file 1e*; *Supplementary file 1l* total sleep time LMM: –0.12 [–0.34, 0.12]; sleep fragmentation LMM: –0.05 [–0.33, 0.23]).

In humans, homeostatic regulation of sleep manifests within as well as between nights: sleep wanes and wakeful activity increases over the course of the sleep period as individuals gradually fulfill their sleep requirements (*Winnebeck et al., 2018*). Baboons, in contrast, did not demonstrate this pattern: their probability of being asleep did not decrease as the night progressed despite exhibiting cyclic

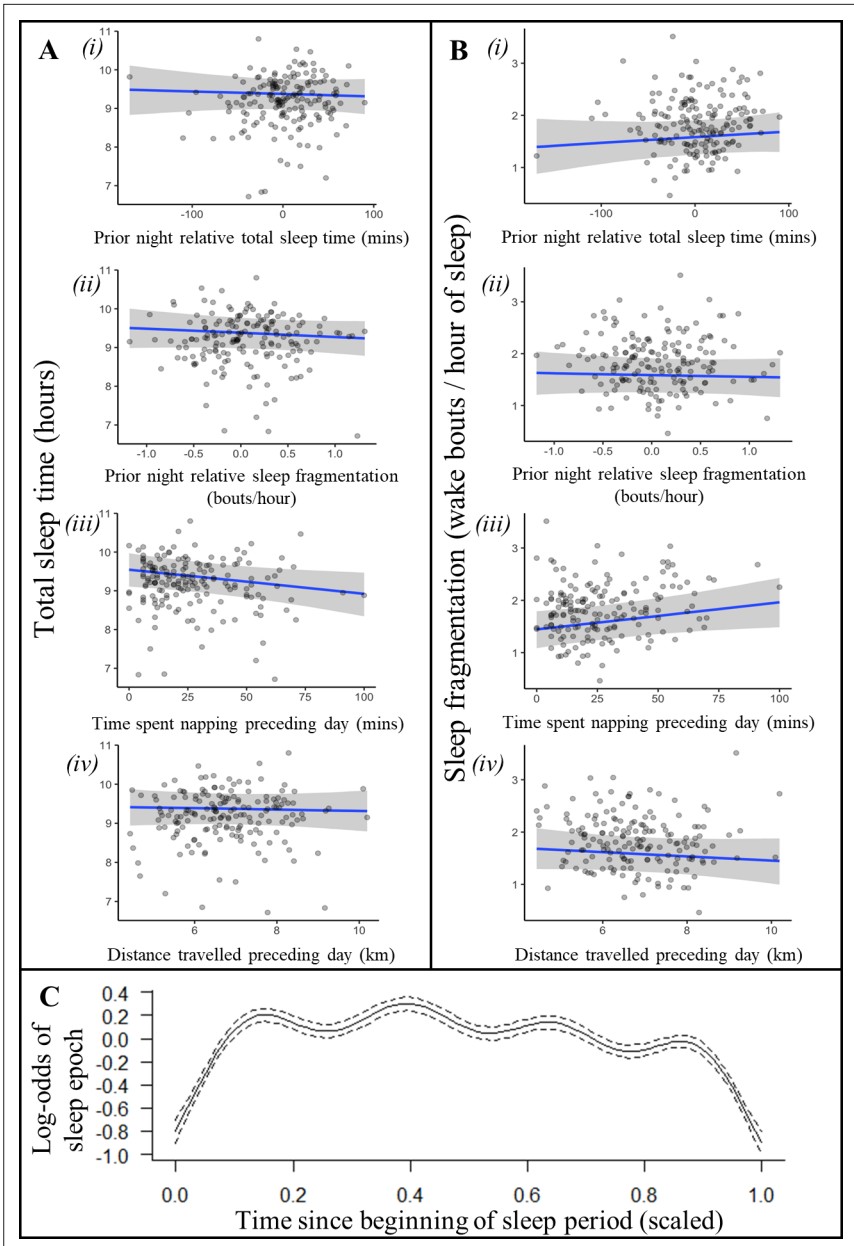

**Figure 2.** Recent history of sleep and activity has a weak influence on baboon sleep patterns. Neither the relative sleep time on the previous night, the relative sleep fragmentation on the previous night, nor the distance traveled on the preceding day influenced sleep duration (**Ai, ii, iv**) or sleep fragmentation (**Bi, ii, iv**), although baboons did sleep less (**Aiii**) and experience more fragmented sleep (**Biii**) following days with more napping. Additionally, the likelihood of a baboon being asleep did not substantially decrease as the night progressed and the baboon payed off its sleep debt (**C**). In (**C**), time since the beginning of the sleep period is scaled from 0 (beginning) to 1 (end of the sleep period). Subplots depict the conditional effect of each variable from models of the data, with raw data points overlaid.

The online version of this article includes the following figure supplement(s) for figure 2:

**Figure supplement 1.** Model output plot of model of total sleep time (for the first 20 days) with all numerical variables standardized.

**Figure supplement 2.** Model output plot of model of sleep fragmentation (for the first 20 days) with all numerical variables standardized.

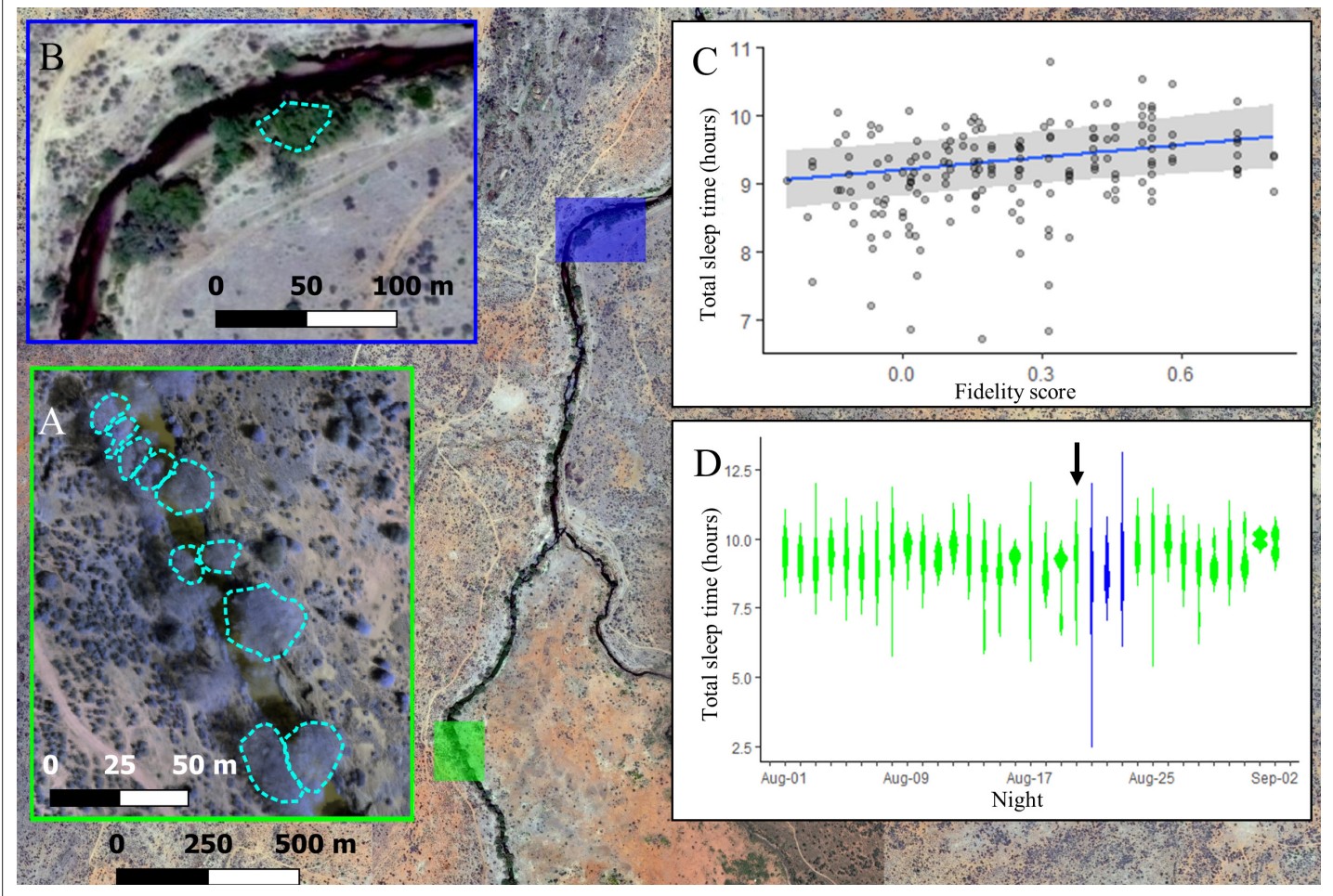

**Figure 3.** The location where baboons sleep has consequences for sleep duration. Group members spent the majority of the study (32/35 nights) sleeping in 10 yellow fever (*V. xanthophloea*) trees in a grove along the Ewaso Ng'iro river (**A**). Within this sleep site, baboons slept longer when sleeping in trees to which they showed high fidelity (**C**). At 20:55 on the 21st night of the study, a leopard mounted an unsuccessful attack on the group in their sleep site. The following day, the baboons moved to a new sleep site 1.5 km away from their main sleep site (**B**). Baboons slept substantially less following this change in sleep site, but this effect did not persist beyond the first night in the new location (**D**). (**C**) Depicts the conditional effect from a model of the data, with raw data points overlaid, and (**D**) depicts a violin plot of the data, with color corresponding to the sleep site (**A**, **B**). The arrow in (**D**) indicates the night on which a leopard launched a failed attack on the group.

The online version of this article includes the following figure supplement(s) for figure 3:

**Figure supplement 1.** Comparison of the Shannon entropies of individuals' sleep tree occupancy within their main sleep site to a null distribution produced by 1000 identity permutations.

**Figure supplement 2.** The conditional effect of tree identity on total sleep time.

**Figure supplement 3.** The conditional effect of night condition on total sleep time.

**Figure supplement 4.** The conditional effect of night condition on sleep fragmentation.

sleep patterns that are otherwise similar to patterns of human sleep (*Figure 2C*; generalized additive mixed model [GAMM]: $r^2_{adj}$ = 0.008, $F_{(8.741)}$ = 89.16, p<1 × 10$^{-15}$).

The location where baboons slept had a strong influence on sleep duration, with individuals experiencing reduced sleep quality in less familiar locations. For the first 21 nights of the study, group members slept at the same site along the Ewaso Ng'iro river, distributed across 10 adjacent yellow fever (*Vachellia xanthophloea*) trees (*Figure 3A*). Individuals showed high fidelity to particular sleep trees (*Figure 3—figure supplement 1*; one-tailed two-sample Kolmogorov–Smirnov test: p<1.0 × 10$^{-9}$), returning each night to one of the small set of available trees populated by the group. Not only did the choice of tree itself influence sleep duration (*Supplementary file 1c, d*, *Figure 3—figure*

supplement 2), but the individual's familiarity with their selected tree impacted how much they slept. Baboons slept longer in trees to which they showed higher fidelity (*Figure 3C*; LMM: 0.21 [0.05, 0.36]), with individuals sleeping up to 33.4 min longer in the tree to which they showed highest fidelity than in the tree to which they showed lowest fidelity. Baboons did not compensate for shorter sleep duration with less fragmented sleep when sleeping in nonpreferred trees (LMM: –0.05 [–0.24, 0.14]).

During the 21st night of the study, we heard the snarls and growls of a large felid, followed by sustained baboon screams and alarm calls that were emitted from the study group's sleep trees at 20:55. After listening to recordings of the vocalizations of the large cats present at the study site, and because leopards are the only predators that readily attack baboons in their sleep site during the night (*Busse, 1980*; *Cheney et al., 2004*), we concluded that the commotion reflected a leopard attack, and that the leopard attack had failed, as all members of the group were present the next morning. On the day following this attack, the group moved to a less commonly used sleep site 1.5 km away (*Figure 3B*). They remained at this sleep site for three nights before returning to sleep at their main sleep site. While the baboons showed no substantial change in their sleep duration or fragmentation on the night of the leopard attack (*Figure 3D*, *Figure 3—figure supplement 3*, *Figure 3—figure supplement 4*; total sleep time LMM: –0.25 [–0.87, 0.39]; sleep fragmentation LMM: –0.32 [–0.96, 0.33]), they slept 72 min less and exhibited 0.50 more wake bouts per hour of sleep, on average, upon moving to the less familiar sleep site (*Figure 3D*, total sleep time LMM: –1.52 [–2.15, –0.86]; sleep fragmentation LMM 0.96 [0.33, 1.60]). This decrease in total sleep time and increase in fragmentation following the change in sites was limited to the first night in the new sleep site, after which sleep duration and fragmentation returned to normal (*Figure 3D*, *Figure 3—figure supplement 3*, *Figure 3—figure supplement 4*).

Sleeping in a social context also impacted sleep patterns. Contrary to predictions of the sentinel hypothesis, the proportion of the night in which at least one individual was awake was significantly less than expected by chance (*Figure 4A*, *Figure 4—figure supplement 1*, Fisher's exact test: p<0.0001; *Figure 4—figure supplement 2B*, Fisher's exact test: p<0.0001), suggesting that, rather than staggering periods of nocturnal wakefulness, group-mates were actually synchronized in their sleep-wake patterns throughout the night. Confirming this synchronization, we found that a significantly greater proportion of the group exhibited the same simultaneous behavior, either being asleep or awake, than expected (*Figure 4B*, *Figure 4—figure supplement 1*, Fisher's exact test: p<0.0001; *Figure 4—figure supplement 2C*, Fisher's exact test: p<0.0001). Moreover, pairs of baboons showed more synchronization when sleeping in the same tree than when sleeping in different trees (*Figure 4C*; LMM: 0.56 [0.47, 0.64]), which suggested that sleeping individuals may awaken in response to the activity of group-mates in their local environment, or that external disruptions in the local environment may simultaneously waken all group members in the vicinity. To distinguish between these potential explanations, we tested the influence of the number of group-mates in an individual's local environment on their total sleep time and found that individuals slept less when sharing their sleeping tree with more group-mates (*Figure 4D*; LMM: –0.53 [–0.87, –0.19]). Each additional tracked group-mate in a tree resulted in a 4.0 min decrease in total sleep time. Individuals also appeared to experience more fragmented sleep when sleeping in the proximity of a greater number of group-mates, but large uncertainty in the model estimate prevents a definitive conclusion about the influence of the social environment on sleep fragmentation (LMM: 0.26 [–0.15, 0.67]).

Building on the evidence that co-sleeping baboons disrupt each other's sleep, we tested whether baboons maintain sleep homeostasis by sleeping deeper, rather than longer, following nights of poor sleep. Because sleep depth is measured by the amount of stimulation needed to awaken an animal (i.e., arousal threshold), we assessed sleep depth in situ by evaluating the probability of an individual waking in response to the activity of a neighboring group member. We found that although individuals were indeed substantially more likely to wake following the awakening of others in their sleep tree, neither shorter duration nor more fragmented sleep the previous night dampened their responsiveness to group-mates (Bernoulli generalized linear mixed model [GLMM]: influence of group-mate activity: 0.25 [0.16, 0.34], interaction between influence of group-mate activity and previous night relative total sleep time: –0.01 [ –0.13, 0.10], interaction between influence of group-mate activity and previous night relative sleep fragmentation: –0.01 [ –0.11, 0.09]).

We found no influence of moon phase or the minimum ambient temperature during the night on baboon sleep duration or fragmentation (total sleep time LMM: moon phase: 0.05 [–0.12, 0.22],

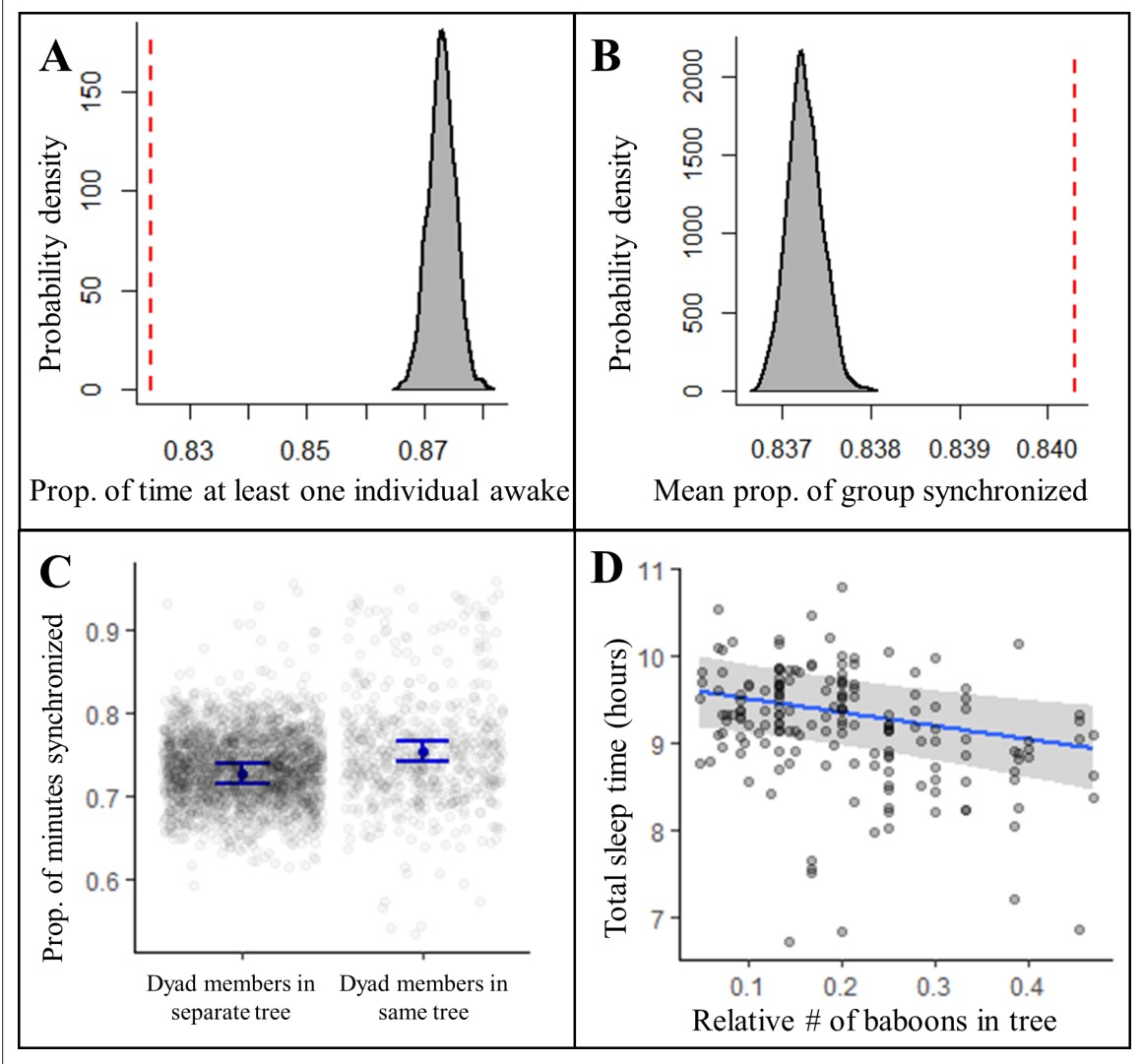

**Figure 4.** Collective dynamics within the sleep site influence sleep patterns. Group-mates' periods of nocturnal wakefulness were not staggered, but rather synchronized, as indicated by a significantly lower proportion of time with at least one individual awake (**A**, dotted red line; Fisher's exact test: p<0.0001) and a significantly greater proportion of the group exhibiting synchronized behaviors (**B**, dotted red line; Fisher's exact test: p<0.0001) than expected based on 1000 time-shifted datasets (gray distribution). Synchronized sleep patterns likely result from individuals waking in response to the nighttime activity of nearby group-mates as dyads show greater synchronization when dyad members sleep in the same tree compared to when they sleep in different trees (**C**). As a consequence of these local social perturbations, baboons sleep less when sleeping in trees with more group-mates (**D**). Subplots (**C**) and (**D**) depict the conditional effects from models of the data, with raw data points overlaid.

The online version of this article includes the following figure supplement(s) for figure 4:

**Figure supplement 1.** A toy example of the procedure we used to test for sentinel behavior and synchronization of nighttime sleep-wake schedules.

**Figure supplement 2.** An alternative permutation procedure used to test for sentinel behavior and synchronization of nighttime sleep-wake schedules, and the results produced by this approach.

temperature: –0.04 [–0.20, 0.13]; sleep fragmentation LMM: moon phase: –0.08 [–0.30, 0.13], temperature: 0.04 [–0.17, 0.26]).

## Discussion

In this study, we demonstrate that the ecological and social demands that animals experience in the natural world can take precedence over the maintenance of sleep homeostasis. We show that, while baboons sleep less in unfamiliar environments, and while their sleep is disrupted by the activity of group-mates, their recent history of sleep and physical exertion has only a limited role in influencing

sleep behavior. Because baboons are highly vulnerable to nocturnal predation (*Busse, 1980*; *Cheney et al., 2004*; *Isbell et al., 2018*) and because they experience fitness benefits from maintaining strong social bonds (*Silk et al., 2009*), sacrificing sleep to maintain alertness in novel environments and to remain close to group-mates may represent critical adaptations. Our results highlight the trade-offs that group-living animals navigate when investing in sleep in the wild, and in doing so, challenge the centrality of the role that homeostasis has played in shaping sleep patterns in the environment in which sleep evolved. Decades of research in the laboratory and at the bedside have implicated homeostatic regulation as a key driver of sleep patterns, with the sleep rebound that follows periods of deficit facilitating the maintenance of a physiologically required amount of sleep (*Amlaner et al., 2009*). However, sleep studies have traditionally investigated sleep in highly controlled environments, where the costs of investing in sleep are largely absent. Our findings suggest that, in the natural world, 'sleep need' may be a relatively flexible concept, with variation in sleep investment driven as much by the opportunity costs of sleep as by its physiological benefits.

There are substantial opportunity costs of devoting a significant portion of every day to sleeping. Sleeping animals are highly vulnerable to predation (*Lima et al., 2005*), and our results suggest that individuals sleep less when the risk of predation is particularly high. Baboon group members showed high fidelity to particular locations within their main sleep site, and individuals sacrificed sleep both when sleeping in nonpreferred trees as well as upon moving to a new, less familiar sleep site following a leopard attack. Given that predation risk tends to be greater in unfamiliar locations (*Forrester et al., 2015*; *Gehr et al., 2020*; *Yoder et al., 2004*), baboons appear to trade sleep for vigilance according to the current risk of predation. Notably, however, we did not find that baboons decreased their investment in sleep on the night of the leopard attack. This surprising result may reflect leopards' disinclination, as stealth hunters, to launch repeated attacks (*Hayward et al., 2006*; *Lin et al., 2020*), or indicate that baboons perceive uncertainty in the level of risk as potentially more dangerous than a confirmed threat. It is also possible that baboons did, in fact, sleep less following the attack, but remained exceptionally still, in a state of highly elevated vigilance. Because accelerometer-based sleep classification performs poorly in distinguishing motionless wakefulness from sleep, this state would likely have been falsely classified as sleep.

Engaging in sleep precludes investment in a variety of behaviors, in addition to antipredator vigilance, that are important to fitness (*Aulsebrook et al., 2016*; *Lesku et al., 2012*; *Lima et al., 2005*; *Roth et al., 2010*). Consistent with our results, recent studies in ecologically relevant contexts have revealed that animals forego sleep when ecological demands increase the associated opportunity costs. For example, while engaging in long, nonstop flights, great frigatebirds reduced the amount they slept by 92.7%, without apparent physiological consequences (*Rattenborg et al., 2016*). Similarly, male pectoral sandpipers greatly reduce their time spent sleeping during their short and intense mating season, and males that slept less actually experienced higher reproductive success (*Lesku et al., 2012*). Thus, across contexts and taxa, ecological pressures appear to supersede investment in sleep in the wild.

Animals might maintain homeostasis in the face of significant opportunity costs of sleep by sleeping more intensely, rather than longer, in response to elevated sleep need. Baboons in this study, however, did not appear to compensate in this way. We reached this conclusion using two distinct approaches of inferring sleep depth from accelerometry data. First, we measured the fragmentation of sleep, which has been shown to correlate inversely with sleep depth (*Bastuji and García-Larrea, 1999*). Second, we directly assessed changes in arousal thresholds by tracking the propensity to wake in response to the nocturnal activity of neighboring group-mates. These approaches offer potentially promising avenues to expand our ability to noninvasively quantify the sleep behavior of animals in their natural habitats using accelerometry. However, validation against polysomnography – the gold standard for recording sleep – should be a priority for future work.

Recent technological advances allowing for the use of polysomnography in field conditions have played an important role in revealing the ecological trade-offs that wild animals face when navigating decisions about when, where, and how to sleep (*Davimes et al., 2018*; *Lesku et al., 2011*; *Lesku et al., 2012*; *Rattenborg et al., 2008*; *Rattenborg et al., 2016*; *Scriba et al., 2013*; *Voirin et al., 2014*). Although these advances hold great promise for wider application in the future, the invasive nature of polysomnography unfortunately limits its current use to taxa whose daily activities do not interfere with electrodes implanted either subdermally or intercranially. Because baboons are highly

dexterous and engage in frequent allogrooming, we were unable to apply this gold standard, and instead, resorted to an alternate method to ask and answer important questions about the ecology of sleep in a wild social primate. Accelerometer-based sleep classification – a tool already used to investigate sleep across terrestrial (human: *Jones et al., 2019*; *Patel et al., 2017*; nonhuman: *Bäckman et al., 2017*; *Davimes et al., 2018*; *Gravett et al., 2017*; *Lesku et al., 2011*; *Malungo et al., 2021*; *Qin et al., 2020*; *Reinhardt et al., 2019*; *Reyes et al., 2021*; *Samson et al., 2018*; *Sellers and Crompton, 2004*; *Sri Kantha and Suzuki, 2006*; *Suzuki et al., 2018*) and marine taxa (*Miller et al., 2008*; *Mitani et al., 2010*; *Wright et al., 2017*) – offered a valid alternative to polysomnography. We note that the use of accelerometry can introduce biases in sleep monitoring, typically by overestimating total sleep time as a result of an inability to distinguish resting wakefulness from sleep (*Ancoli-Israel et al., 2003*; *de Souza et al., 2003*). Accelerometry is also limited in its ability to differentiate the stages of sleep (*Conradt et al., 1997*, but see *Devine et al., 2020*; *Winnebeck et al., 2018*). However, if these biases and limitations are considered during the interpretation of results, the use of accelerometry to investigate sleep provides an immediate opportunity to shed light on how diverse species balance their physiological sleep requirements with ecological opportunity costs that vary according to natural history, trophic level, community composition, climate, and local environment. Further, the relative ease of accelerometer deployment, and its prevalence in ecological research, changes the scale at which sleep behavior can be studied, enabling the simultaneous and long-term monitoring of sleep at the population level. This rescaling of sleep research creates many new opportunities, one of which is the ability to record sleep in the majority of social group members and thus explore an exciting new scientific frontier: the collective dynamics of sleep.

Using accelerometry to track the sleep patterns of nearly an entire group of wild baboons, we demonstrated the importance of the social environment in shaping the sleep patterns of group-living animals. Contrary to the predictions of the sentinel hypothesis (*Samson et al., 2017*; *Snyder, 1966*), periods of nocturnal wakefulness of group members were not staggered, but rather synchronized, particularly with nearby group-mates. Baboons also slept less when in close proximity to a greater number of group-mates. Taken together, these results suggest that group-mates disrupt each other's sleep. Social disruptions may result from group-mates actively interacting with each other during the night. Gregarious animals often invest substantially in building and maintaining social relationships with their group-mates (*Ward and Webster, 2016*), and these bonds can prove essential to their fitness (*Cameron et al., 2009*; *Campos et al., 2020*; *Frère et al., 2010*; *Riehl and Strong, 2018*; *Silk et al., 2009*). Because animals have limited time to devote to maintaining their social bonds during the day (*Dunbar, 1992*), they may actively sacrifice sleep in order to invest in these relationships at night. Alternatively, social animals may wake in response to the periodic waking and repositioning of their group-mates during the night, and thus, socially disrupted sleep is, perhaps, an inherent by-product of sleeping in a group. Simply remaining in a cohesive group may therefore present a challenge to obtaining sufficient sleep.

Social animals may jeopardize sleep homeostasis to maintain cohesion with their conspecifics because remaining in close proximity to their group-mates during the sleep period could prove essential to their fitness. Individuals likely benefit from the dilution of predation risk that is achieved through group cohesion, particularly when they are sleeping and thus highly vulnerable to predators (*Lehtonen and Jaatinen, 2016*). Collective vigilance may also reduce the risk of predation for group members. Even in the absence of collective vigilance optimization via nonrandomly staggered wakefulness, the proportion of the night with at least one group member awake is still likely to be substantially greater than any particular individual's investment in vigilance. In our study, at least one individual in the group was awake for 394 ± 11 min (82% ± 2%) from 21:00 to 05:00, although each individual was only awake for 79 ± 1 min (16% ± 0.2%) of the same period. *Samson et al., 2017* found high levels of collective vigilance during the night in a group of Hadza hunter-gatherers, and they suggest that this collective vigilance may facilitate higher-intensity sleep (*Samson and Nunn, 2015*). Future studies leveraging the indices of sleep depth that we applied in this study or advances in polysomnography (i.e., EEG) that may eventually allow its application in wild social animals could enable a test of this possibility.

Unexpectedly, we found that adult baboons slept longer than subadults and juveniles, and males slept longer than females. This contrasts with previous research that found age differences in sleep patterns linked to physiological demands during the development of the central nervous system (*Amlaner et al., 2009*) and sex differences in sleep tied to the influence of sex steroids (*Mong and*

Cusmano, 2016), with younger individuals sleeping more than older individuals (Knutson, 2014; Ohayon et al., 2004; Olds et al., 2010; Steinmeyer et al., 2010; Stuber et al., 2015) and females sleeping more than males in birds and humans (Lendrem, 1983; Mong and Cusmano, 2016; Roehrs et al., 2006; Steinmeyer et al., 2010; Stuber et al., 2015). Our surprising results here may have been caused by biases in our study sample. The bio-logging collars were too heavy to mount on young juveniles and infants – the individuals that we would have predicted to sleep the most due to their rapidly developing central nervous system (Frank, 2020; Landolt and Dijk, 2019). Thus, there may have been an overall decrease in time spent sleeping with age that we were unable to observe because we did not collect data on the youngest individuals. This result may also be an artifact of the tendency of accelerometer-based sleep monitoring to classify resting wakefulness as sleep (Supplementary file 1r; Ancoli-Israel et al., 2003; de Souza et al., 2003). Older individuals may rest quietly when waking during the night, thus falsely determined to be asleep according to their accelerometry, whereas younger individuals may be more likely to resume activity upon waking. However, if these findings are not the result of a bias in our study subject inclusion or sleep recording technique, they may reflect variation in the vulnerability to predation among the age-sex classes in this highly sexually dimorphic species (Cheney et al., 2004), with young and female baboons likely realizing a higher cost of sleep than adult males. Individuals may also differ in their sleep patterns due to their ability to gain access to a high-quality sleep location within the group's sleep site. Our results have demonstrated the importance of location to sleep. However, group-mates may differ in their access to preferred sleep locations, particularly if preferred locations are limited. Baboon groups are structured by linear dominance hierarchies that shape the priority of access to resources (Cheney and Seyfarth, 2008; King et al., 2009; Marshall et al., 2015), and individuals can leverage their affiliative and kinship relationships to obtain resources that they would not be able to access based on social rank alone (Sick et al., 2014). Further research is needed to investigate the extent to which these complex social dynamics influence an individual's ability to obtain a preferred sleep location and, thus, a good night's sleep.

In addition to highlighting social dynamics as a key driver of sleep patterns in group-living species, our study provides important insights into selective pressures that may have shaped the evolution of human sleep. The physiological requirements for sleep and the homeostatic mechanisms that ensure this requirement is fulfilled have long been assumed to be the key drivers influencing the way that our sleep has evolved and the characteristics of our sleep today. However, we suggest that the criticality of homeostatic control in shaping our sleep patterns could be an artefact of sleeping in an environment devoid of the ecological and social costs that sleep would have presented our ancestors. Evidence suggests that, like baboons, early hominins were extremely vulnerable to nighttime predation in their dry savannah habitats (Brain, 1983; Treves and Palmqvist, 2007; Wrangham and Carmody, 2010). Hominins likely remained vulnerable to nocturnal predation until they began to manipulate fire, around which they could sleep to reduce the risk of predation (Samson and Nunn, 2015), and some characteristics of our sleep today may be best explained in light of the vulnerability that sleep imparted on our ancestors. For example, modern humans exhibit decreased sleep quality when sleeping in an unfamiliar environment (Tamaki et al., 2016), similar to the baboons in our study. The lower quality sleep resulting from this aptly named 'first night effect' is limited to the first night in a new location (Tamaki et al., 2016), and our findings suggest that the first night effect may be conserved from an environment where this first night would have been accompanied by poor information about risk, and thus, a higher likelihood of predation. Early hominins would have also experienced a social opportunity cost of sleep as they likely slept in groups (Samson and Nunn, 2015; Willems and van Schaik, 2017) and would have experienced constraints on the time available to maintain their social network during the day until developing the advanced cognition that enabled a more efficient use of time (Nunn and Samson, 2018; Samson and Nunn, 2015). While our sleep has likely evolved substantially from that of our earliest ancestors, with modern human sleep being extremely short and intense compared to that of other primates (Nunn et al., 2016; Nunn and Samson, 2018), a full understanding of the way we sleep involves considering not only the physiological benefits of sleep, but also its ecological and social costs in the environment in which it evolved.

## Materials and methods
### Data collection
We monitored sleep and activity patterns in a group of olive baboons at Mpala Research Centre (MRC), a 200 km² conservancy located on the Laikipia Plateau in Central Kenya. We trapped and anesthetized 26 individuals, which comprised more than 80% of the adults and subadults in the study group (see *Strandburg-Peshkin et al., 2015* for details on capture methodology). Upon capture, we noted the age class and sex of each baboon, as well as whether the baboon was lactating. We fit each individual with a GPS and accelerometry collar that recorded the baboon's GPS location at 1 Hz sampling interval and continuous triaxial accelerations at 12 Hz/axis from 06:00 to 18:00. From 18:00 to 06:00, the collars recorded a 2.5 s burst of accelerations at 10 Hz/axis at the beginning of every minute. The collars were programmed to collect data from August 1, 2012, to September 6, 2012, but due to a programming glitch, several collars stopped collecting data prematurely (*Supplementary file 1a*). In total, we collected 483 days of GPS data and 506 nights of accelerometry data. We also collected high-resolution drone imagery of the group's most commonly used sleep site (see *Strandburg-Peshkin et al., 2017* for details).

While downloading data from the collars during the night of August 20, 2012, we heard the snarls and growls of a large felid, followed by sustained baboon screams and alarm calls that were emitted from the study group's sleep site at 20:55. Upon comparison of the vocalizations to recordings of felids at the study site, we found that the vocalizations were most similar to the those of a leopard. Leopards are also the primary predator of baboons, and the only felid predator that readily attacks baboons in their sleep trees (*Busse, 1980*; *Cheney et al., 2004*; *Cowlishaw, 1994*). We therefore concluded that the vocalizations were emitted by a leopard that had launched an attack on the group. Although we were only 45 m away from the nearest sleep tree, we were unable to make any direct visual observations of either the predator or the baboons during the attack due to the low-light conditions (sunset had occurred at 18:41). The following morning, a group count revealed that all group members were present and alive. We therefore concluded that the attack had failed.

### Sleep analysis
We used the accelerometry data to classify sleep behavior by adapting a method presented in *van Hees et al., 2018* that was developed for extracting metrics of sleep in humans from wearable accelerometry devices. The process of determining the sleep period, defined as the period from sleep onset to waking, is summarized in *Figure 5*.

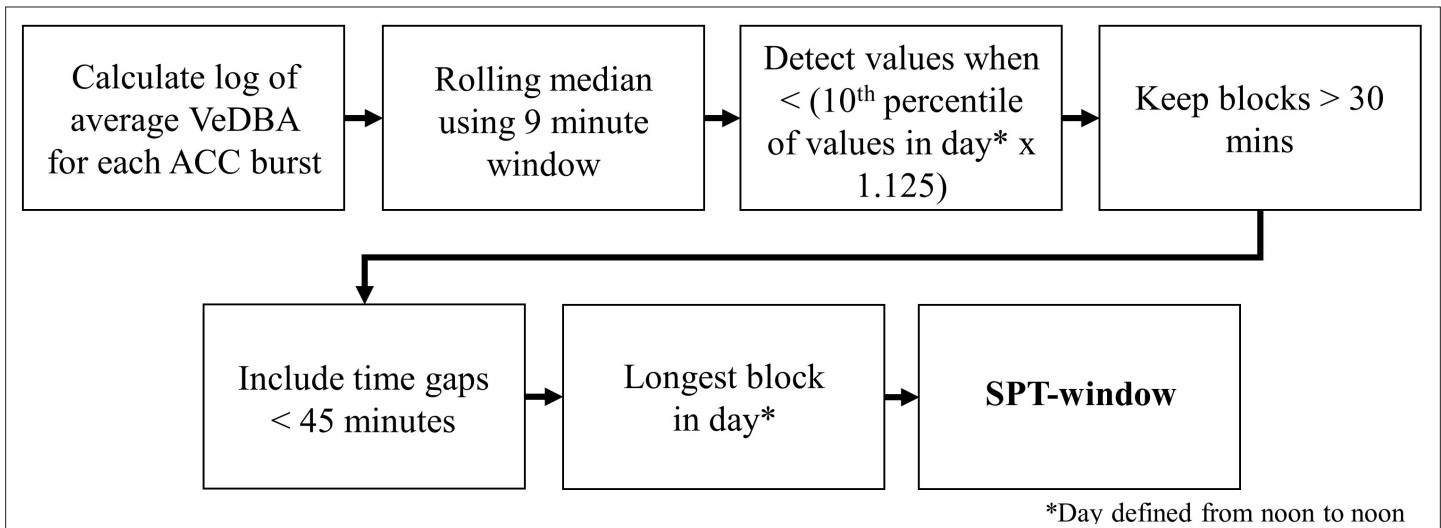

**Figure 5.** SPT-window detection algorithm adapted from Figure 1 in *van Hees et al., 2018*.

The online version of this article includes the following figure supplement(s) for figure 5:

**Figure supplement 1.** Examples of the three different behaviors, 'wakefulness,' 'resting wakefulness,' and 'sleep,' that were scored during the validation study.

To uniformize the accelerometry sampling schedule, we downsampled and interpolated the daytime accelerometry data such that it matched the 10 Hz bursts of accelerometry collected during the night. We calculated the VeDBA using a 0.7 s time window and generated the log of the average VeDBA for the 2.5 s burst each minute. We then calculated a rolling median of the log VeDBA with a 9 min window. Following *van Hees et al., 2018*, continuous periods of at least 30 min during which the rolling median of the log VeDBA was less than the 10th percentile of the log VeDBA multiplied by 1.125 were considered sleep blocks. Any blocks within 45 min of each other were merged into sleep periods. If this resulted in more than one sleep period, the longest sleep period in the day, defined as a noon-to-noon period, was considered the sleep period. The beginning and end of the sleep period represents the time of sleep onset and waking, respectively. Of the 506 baboon-nights of accelerometry data, we successfully calculated the sleep period for 491 baboon-nights.

Adapting the method developed by *van Hees et al., 2015*, we classified each minute epoch both within and outside of the sleep period as representing either sleep or waking behavior. As above, the 10th percentile of the log VeDBA multiplied by 1.125 served as the classification threshold, and we classified epochs as indicating sleep when the log VeDBA for at least three consecutive epochs was below the log VeDBA threshold value. We classified all other epochs as representing waking behavior. Consistent with previous sleep analyses, we measured total sleep time as the total number of minutes of sleep epochs during the sleep period. We measured sleep efficiency as the total sleep time divided by the duration of the sleep period. We calculated sleep fragmentation as the total number of distinct wake bouts (i.e., separated by at least three epochs of sleep) within the sleep period that were greater than or equal to 2 min in duration, divided by the total sleep time, following *Samson and Nunn, 2015*. We measured sleep time during the day – time spent napping – as the number of minutes of sleep epochs from 07:30 to 17:30 as these times were reliably within the waking period (*Figure 1C*) and using standardized times prevented a spurious negative correlation between time spent sleeping during the waking period and total sleep time during the sleep period that would result from the waking period prior to or following short sleep periods having a greater number of potential epochs that could be considered sleep.

The accelerometer units occasionally failed to collect data according to their programmed sampling schedule. Because insufficient data in a given day would prevent a reliable calculation of the threshold value for the sleep classification and produce variability in the number of potential sleep epochs, we did not include data for the sleep period time, sleep onset time, waking time, total sleep time, sleep efficiency, sleep fragmentation, or napping time (both on the prior day and following day) from noon-to-noon periods missing at least 120 (8.3%) accelerometry bursts, which decreased the number of baboon-nights from 491 to 368. We further removed data for the sleep period time, sleep onset time, waking time, total sleep time, sleep efficiency, and sleep fragmentation from noon-to-noon periods missing at least 20 consecutive accelerometry bursts as the determination of the sleep period is sensitive to gaps between consecutive accelerometry bursts, resulting in a final number of 354 sleep periods analyzed. We did not remove data for napping time on these days because measuring napping time did not depend on the determination of the sleep period.

## Validation of sleep classification algorithm

The algorithm from which the sleep classification technique is adapted is well-validated using polysomnography (C-statistic = 0.83–0.86) to both classify sleep behavior and determine the sleep period in humans (*van Hees et al., 2015*; *van Hees et al., 2018*). Although the classification of sleep in nonhuman primates using devices and algorithms that were validated with polysomnography only in humans has become a common practice in sleep research (*Barrett et al., 2009*; *Brutcher and Nader, 2013*; *Reinhardt et al., 2019*; *Reyes et al., 2021*; *Samson et al., 2018*; *Sri Kantha and Suzuki, 2006*; *Zhdanova et al., 2002*), we returned to the study site in July 2019 to validate the accelerometer-based sleep classification. Because logistical and ethical limitations prevent the use of polysomnography in free-ranging, highly dexterous animals, we compared the accelerometer-based sleep classification to direct observations of wakeful and sleeping baboons fit with accelerometer collars for validation (*Figure 5—figure supplement 1*), as suggested by *Rattenborg et al., 2017*. Behavioral observations were facilitated by high-resolution thermal imagery (FLIR T1020, FLIR Systems Inc, Wilsonville, OR). The validation study determined that our accelerometry-based classification of sleep exhibits 80.7% accuracy (*Supplementary file 1r*; see Sleep Validation Study for further details of validation study).

## Physical activity

Using the GPS data, we calculated each individual's daily travel distance. To avoid accumulation of GPS positional error overestimating the actual daily travel distance, we calculated daily travel distance only after discretizing the GPS data to 5 m resolution (*Strandburg-Peshkin et al., 2017*). We removed travel distance data on days on which a baboon's GPS collar first began taking fixes later than 07:30 or took its last fix before 17:00. Between these times, the group was often on the move, and thus delayed onset and premature offset of GPS devices that infringed upon this period would likely underestimate travel distances. We further removed one individual's data from the first half of the study due to a temporary collar malorientation that resulted in exaggerated GPS error.

We also calculated cumulative activity during the day from the accelerometry data. Using the continuous 12 Hz accelerometry data, we calculated VeDBA from 06:00 to 18:00 using a 0.5 s time window, averaged VeDBA over each minute, and then summed these values to generate a cumulative measure of activity during the day.

## Sleep location characterization and fidelity

Visualization of the GPS data indicated that individuals remained reliably stationary until at least 06:15 every day, and thus we determined the location in which each baboon slept from the median of the first 10 GPS locations that occurred before 06:15. If an individual's GPS collar did not successfully collect 10 locations before 06:15, its data on this day were excluded from analyses involving sleep location. This resulted in the removal of 9/483 baboon-days of data. In ArcGIS, drone imagery was used to trace the crowns of distinct sleep trees within the group's main sleep site. We determined that an individual slept in a particular tree if its sleep location was within the traced polygon of that tree crown. Sleep locations that fell outside the crown of a tree, likely reflecting minor error in the GPS location estimates, were assigned to the closest sleeping tree. Only 32/469 sleep locations (6.8%) had to be assigned to a sleep tree in this manner. In rare cases where an individual's sleep location was greater than 10 m from the crown of the closest sleep tree (5/474 cases – 1.1% of baboon-days), its data on this day were excluded from analysis.

Analysis of the sleep location data revealed that, over the course of the study, the baboons slept in two distinct sleep sites that were separated by approximately 1.5 km. The group slept at their main sleep site for the first 21 nights of the study, and then spent 3 nights in a different sleep site after the failed leopard attack on the 21st study night. The group then returned to the main site for the duration of the study. In total, they spent 32/35 (91.4%) nights at their main site and 3/35 (8.6%) nights at the alternate sleep site. While the entire group slept in a single tree at the less frequently used sleep site, the group's main sleep site contained 10 trees across which the group slept. We performed a permutation test to investigate whether individuals exhibited consistency in the tree in which they chose to sleep. We calculated the Shannon entropy of each individual's sleep tree usage and compared these Shannon entropies to those produced from each of the 1000 random exchanges of the locations of individuals on each night. Permuted values provided a null distribution controlling for potential sleep tree usage as the distribution of individuals across the sleep trees each night from the empirical data was maintained in the permuted data. Shannon entropy is a measure of the uncertainty of a random variable and is given by the following equation:

$$H(X) = -\sum_{X=x} p(x) \, log(p(x))$$

Thus, a lower Shannon entropy in the empirical data compared to the permuted data in this context would signal sleep tree fidelity, with an individual sleeping more often in certain trees and less often in other trees than expected by chance. To determine whether the baboons exhibited significant sleep tree fidelity, we compared the distribution of the group's empirical entropies to the distribution of entropies produced from the permutations with a one-tailed two-sample Kolmogorov–Smirnov test. As determining fidelity requires several nights of data, we did not include entropy values, either empirical or permuted, from individuals with less than four nights of data. We also limited this analysis of tree fidelity to the first 14 days of data as the number of individuals on which we have data decreases sharply after this day (*Supplementary file 1a*), which decreases the possible permutations.

After determining that individuals showed nonrandom sleep tree selection (see Results), we then calculated an individual-specific fidelity index for each tree. This fidelity index was measured as the average number of nights an individual slept in a particular tree in the 1000 permutations subtracted

from the number of nights the individual actually slept in that particular tree. Again, we did not calculate fidelity indices for individuals with less than four nights of data.

## Pattern of sleep-wake behavior across the group

We tested whether individuals staggered their periods of nocturnal wakefulness or, conversely, synchronized them beyond the level expected by chance. For this analysis, we subset the data to times between 21:00 and 05:00 as these times consistently fell within the bounds of the sleep period of all individuals. We calculated the proportion of minute epochs across all nights in which at least one group member was awake and the proportion of the group that was synchronized in their behavior (either sleep or wakefulness) during each minute epoch, averaging across all epochs. We then calculated these same proportions, but after applying a random time shift to each individual's time series of sleep-wake epochs on each night (*Figure 4—figure supplement 1*). We repeated this procedure 1000 times to develop a null distribution of the proportion of epochs during the night in which at least one individual is awake and a null distribution of the average proportion of the group that was synchronized, and we compared the empirical proportions to their respective null distributions statistically with a Fisher's exact test. The p-value thus represents the proportion of time-shifted values that were as extreme or more extreme than the empirical value. Shifting the data in time rather than permuting it allowed us to develop null distributions while maintaining the autocorrelation structure of the data.

To confirm the robustness of our findings, we again tested for collective vigilance and synchronization, comparing the empirical values defined above to null distributions produced using an alternative method. In this method, rather than applying a random time shift to each night of each individual's data, we maintained the real time associated with the time series data, but we permuted the night associated with each time series (*Figure 4—figure supplement 2*). This permutation method controlled for the possibility that baboons exhibited synchronized sleep patterns simply due to a stereotyped schedule of activity that happened to be consistent across baboons and across nights. We compared empirical values to the null distributions created by these night permutations with a Fisher's exact test.

## Statistical analysis of sleep

Data were processed using the statistical analysis software R version 4.0.5 (*R Development Core Team, 2021*). We only included the first 20 study nights in the analyses of sleep, except where specified, due to concerns that the leopard attack that occurred on the 21st night could potentially disrupt typical sleep patterns. To compare the effects of various physiological, ecological, and social predictors of sleep, we modeled total sleep time and sleep fragmentation with Bayesian LMMs, with random intercepts for individual identity and night, and fixed effects of age, sex, distance traveled in the preceding day, napping time during the preceding day, relative time spent sleeping the previous night, relative sleep fragmentation the previous night, identity of the sleep tree, fidelity index for the current sleep tree, relative number of individuals in the sleep tree, phase of the moon, and minimum ambient temperature during the night. We created a separate model that included cumulative daytime VeDBA instead of distance traveled because cumulative daytime VeDBA was highly correlated with distance traveled during the day. An individual's relative time spent sleeping the previous night was measured as the difference between its total sleep time on the previous night and its average total sleep time. This relative measure controlled for positive correlations between total sleep time on the previous night and current night total sleep time that would result purely from among-individual variation in total sleep time – a scenario that would not be sufficiently controlled for by the individual identity random effect in this model. An individual's relative sleep fragmentation on the previous night was calculated for the same motivation. We calculated the relative number of individuals in the sleep tree by dividing the number of individuals in the sleep tree by the total number of individuals who were successfully assigned to a sleep tree on that given night, to control for the decrease in the number of individuals in each sleep tree over the course of the study that resulted from premature termination of data collection in several collars. Moon phase was a continuous variable realizing values from 0 to 1 (with 0 representing a new moon and 1 representing a full moon), and we collected this data for the days of the study using the 'sunCalc' package in R (*Benoit and Elmarhraoui, 2019*). The minimum ambient temperature represented the minimum temperature at the sleep site during the

night, determined using interpolated ECMWF air temperature (2 m above ground) data obtained with the Env-DATA functionality (*Dodge et al., 2013*) provided on Movebank data repository (https://www.movebank.org/cms/movebank-main). We standardized all response and predictor variables to allow for comparison of effect sizes among variables. To increase the interpretability of the total sleep time and sleep fragmentation models, we reran the models without standardized variables. Effect sizes reported in the main text are derived from the standardized models, whereas figures produced in the main text, as well as the interpretation of the effect of each variable on the unstandardized sleep time and sleep fragmentation, are derived from the models with unstandardized variables.

To examine the effect of the leopard attack and subsequent sleep site change on sleep parameters, we modeled the effect of particular nights on sleep parameters with Bayesian LMMs. Specifically, we divided data into the following categories: all nights before the leopard attack, the night of the leopard attack, the first night in the new sleep site (i.e., the night following the leopard attack), the second night in the new sleep site, the third night in the new sleep site, and the remainder of study nights, during which the group slept in its main sleep site. Aside from this categorical night variable, we also included age, sex, distance traveled in the preceding day, napping time during the preceding day, relative time spent sleeping the previous night, relative sleep fragmentation the previous night, phase of the moon, and minimum ambient temperature as fixed effects in the models with random intercepts for individual identity. In these models, we did not include sleep tree identity, number of individuals in the sleep tree, and sleep tree fidelity score as the entire group slept in a single tree in the less commonly used sleep site.

We further tested for the effect of prior sleep debt on sleep behavior by modeling the effect of total sleep time and sleep fragmentation on time spent napping the following day. We modeled this relationship with a Bayesian LMM using individual identity and day as random intercepts. We also assessed how the likelihood of sleep progressed through the night. We used a GAMM to model the log-odds of a baboon being asleep in a given epoch as a function of the duration of that epoch from the beginning of the sleep period, scaled such that 0 represents the beginning of the sleep period and 1 represents the end of the sleep period. We included individual identity and night as random intercepts, and to account for autocorrelation in the response variable, we also included an AR1 term in the model.

We tested whether individuals showed higher synchronization of their sleep-wake patterns when sharing the same sleep tree than when inhabiting different trees. With a Bayesian LMM, we modeled the synchronization score between dyads on each night, calculated as the number of minutes from 21:00 to 05:00 in which members of the dyad exhibited the same behavior divided by the total number of minutes in which both individuals had data. We included a binary predictor variable indicating whether dyad members were in the same tree as the only fixed effect variable, and night, the identity of both individuals in the dyad, as well as the identity of the dyad as random intercept variables.

To assess whether baboons experience deeper sleep following nights of poor sleep, we analyzed baboons' arousal thresholds as a function of their total sleep time and sleep fragmentation on the previous night. We leveraged the results of our previous analyses, which suggested that baboons may wake in response to the nighttime activity of their group-mates, and tested how their response to the nocturnal awakening of group-mates in their sleep tree is modulated by their previous night's sleep. We used a Bayesian GLMM of family Bernoulli to model the log-odds of a focal baboon being awake in a given epoch as a function of a binary variable indicating whether any group-mate in the focal baboon's sleep tree was awake in the previous epoch, the focal baboon's relative total sleep time on the previous night, the focal baboon's relative sleep fragmentation on the previous night, and interactions between each of the previous night sleep variables and the binary variable indicating group-mate activity. We included random intercepts for individual identity and night. Because this analysis depended on knowing the location and sleep-wake state of group-mates and because many of the collars ceased collecting data after the 14th day of data collection, we limited this analysis to the first 14 nights of data collection. We only analyzed epochs between 21:00 and 05:00 as we were aiming to assess arousal thresholds well within the sleep period rather than those at onset or during waking. Lastly, we further subset the data to be modeled to the epochs in which the focal baboon had been asleep for the three preceding epochs as we were only interested in the response of sleeping baboons, and not baboons that were already awake, to external stimuli (i.e., their group-mates, in this case).

We carried out all Bayesian analyses with the 'brms' package in R (*Bürkner, 2017*). We used diffuse, mean-zero Gaussian priors for all predictor variables. Model estimates are based off of four independent Hamiltonian Monte Carlo chains with 5000 iterations, 2500 of which were burn-in iterations. Trace plots indicated that mixing was sufficient and that the four chains converged on the same posterior region. Model estimates reported in the text represent the mean of the posterior distribution, along with the lower and upper 95% credible interval bounds from the standardized models. We used package 'mgcv' in R (*Wood, 2011*) to fit the GAMM to the sleep epoch data using a thin plate spline smoothing term with 10 knots.

## Evaluation of potential biases

In this section, we evaluate potential biases and limitations to the generalizability of our results utilizing the STRANGE framework (*Webster and Rutz, 2020*).

### Social background

Because all study subjects were free-ranging, wild baboons living in a group with a size and sex ratio that is typical for the species (study group size vs. typical group size: 46 vs. 15 – 100 [*Ray and Sapolsky, 1992*], study group sex ratio vs. typical sex ratio (males:female): 3:5 vs. 1:2 [*Barton et al., 1996*]), we do not believe that the social background of the study individuals limits the generalizability of our findings to other baboon groups. Because we trapped and collared ~80% (24/29) of the adults and subadults in the study group, we believe that baboons with a wide variety of social statuses are represented in our data, and thus the results generalize well to other groups of baboons.

### Trapability and self-selection

We chose the study group based on the group's size, their proximity to an ideal trapping location, our ability to remotely download collar data from an infrequently used road near their main sleep site, and our avoidance of territorial conflicts with resident primatologists. Although we aimed to study a group that was relatively small, the study group was only slightly, if at all, smaller than other groups in the area. Thus, we do not feel that these selection criteria biased our results. Because we needed to trap individuals before collaring them, bold individuals, which tend to be more likely to enter traps (reviewed in *Biro and Dingemanse, 2009*), may be overrepresented in our data. To mitigate this bias, we manually triggered traps such that we could capture individuals that were hesitant to enter traps and may have only entered the traps once, rather than repeatedly capturing individuals that continuously entered traps. Capturing a large majority of adults and subadults in the group also helped to ensure representation of a wide range of behavioral types. Despite these mitigating measures, the baboons that we were unable to capture likely represented a nonrandom sample. However, research on behavioral syndromes suggests that these presumably shy individuals also tend to be more reactive to external stimuli (*Réale et al., 2010*). We therefore predict that the individuals that we did not capture may have shown even more pronounced social and risk-related disruptions of sleep than those individuals included in our data, and thus, our conclusions should be unaffected by trapability bias. We were able to recover all data collected by the collars, and therefore, data recovery did not create any bias.

Although not trapability bias per se, we did not capture and collar any young juveniles or infants as they were too small to carry the GPS/accelerometry collars. This bias certainly could have contributed to our surprising finding that adults slept more than younger individuals as the youngest individuals, which would have been predicted to sleep the most, were excluded from our study.

### Rearing history

All study individuals were wild, and because they were only semi-habituated, they had relatively limited exposure to humans.

### Acclimation and habituation

Abnormal proximity to humans shortly prior to and during trapping, as well as handling and sedation directly associated with collaring, may have induced high levels of stress for study subjects. Because stress influences sleep (*Han et al., 2012*; *Sanford et al., 2015*), our procedure could have theoretically biased our results. However, our observations of baboons before and after trapping (both during

this study and beyond) suggest that they continue normal behavior very shortly after trapping, and that they acclimate to the bio-logging collars almost instantly. Nonetheless, to ensure that study animals had acclimated to their collars and resumed normal behavior before commencing data collection, we delayed the start of data collection until 3 days after the last baboon had been trapped and collared. The majority of study individuals (~70%) were collared at least 9 days before data collection commenced (*Supplementary file 1a*). Thus, we do not believe that study procedures biased our results. In confirmation, sleep patterns in the first few days of data collection were not abnormal (*Figure 3D*).

### Natural change in responsiveness

Because our study took place near the equator, at a site with relatively little seasonality (*Isbell et al., 2017*), we believe that results from our data collection period are generalizable to other months. We explicitly considered and analyzed the influence of life stage, as well as the influence of the lunar cycle, on sleep patterns.

### Genetic make-up

We have no reason to suspect that the genetic make-up of the study individuals would bias their sleep patterns in comparison to other populations of olive baboons.

### Experience

All study individuals experienced trapping and collaring procedures for the first time during this study, and thus, were equally naïve. See 'Acclimation and habituation' for discussion of the potential for experience with humans during collaring procedures to influence sleep patterns.

## Sleep validation study

To evaluate whether the accelerometer-based sleep classification technique was accurately monitoring sleep in baboons, we returned to MRC in July 2019 to perform a validation study in which we compared the results of the accelerometer-based sleep classification to direct observations of awake and sleeping baboons. Using the procedures described in *Strandburg-Peshkin et al., 2015*, we trapped and anesthetized 27 members of a group of habituated olive baboons, fitting each with a GPS and accelerometry collar. Of the 27 collars deployed, 11 recorded continuous triaxial accelerations at 12 Hz/axis from 06:30 to 18:00 and 0.71 s bursts of accelerations at 56.2 Hz/axis at the beginning of every minute from 18:00 to 06:30. Accelerometry data was collected by each of these 11 collars for up to 31 days. The remaining 16 collars did not collect accelerometry data from 06:30 to 18:00, and thus we excluded data from these collars from the validation study.

We downsampled and interpolated the accelerometry data such that it matched the sampling frequency and schedule of the data collected in 2012 (i.e., the data analyzed for this article). We then applied the sleep classification algorithm described in the Materials and methods to this validation dataset.

To validate the sleep classification algorithm, we performed direct behavioral observations of the baboons at their primary sleep site. We recorded the behavior of the study baboons starting when they approached their sleep site using a FLIR T1020 high-resolution infrared camera (FLIR Systems Inc). Recordings continued into the night for as long as the camera battery allowed (average recording duration [range of recording durations]: 7.4 hr [1.7–14.9 hr]), and we collected thermal imaging data on 21 nights. We identified individuals in the thermal imagery both in real time, via observer narration of the recorded imagery, and post recording by matching movements of individuals in the thermal imagery to the GPS tracks of collared individuals.

Following initial data collection, we used the commercial software Loopy (Loopbio GmbH, Austria) to score the behavior of identified individuals in the thermal imagery. Individuals' behavior was scored as 'wakefulness,' 'resting wakefulness,' or 'sleep' (*Figure 5—figure supplement 1*). Wakefulness refers to any behavior involving active movement (i.e., walking, running) or engaged activity (i.e., allogrooming), whereas resting wakefulness refers to behaviors that are dormant (i.e., sitting), but not in the typical sleeping posture of a baboon (sitting or lying with neck relaxed and head hung). Sustained dormant behavior in the typical sleep posture was considered sleep. Video scoring resulted in a total of 8.0 hr of behavioral observation across a total of 16 individual baboons.

Synchronizing the thermal imagery data with the accelerometry data produced a validation dataset of 301 minute epochs across seven baboons that were both classified as either sleep or wakeful behavior from accelerometry, and scored as wakefulness, resting wakefulness, or sleep from direct observation. With both wakefulness and resting wakefulness representing wakeful behavior, the accelerometer-based sleep classification exhibited an accuracy of 80.7% (*Supplementary file 1r*). Consistent with previous validation studies of the use of accelerometry in measuring sleep (*Ancoli-Israel et al., 2003*; *de Souza et al., 2003*), we found that accelerometer-based sleep classification has difficulty distinguishing resting wakefulness from sleep, and we consider this limitation in our interpretation of the results.

## Acknowledgements

We are grateful to the Kenya National Science and Technology Council (NCTS/RCD/12B/012/26B, NACOSTI/P/15/5727/4608, NACOSTI/P/19/55517/24299, NACOSTI/P/20/1407), Kenyan Wildlife Service (KWS/BRM/5001), and Mpala Research Centre for permission to conduct research. Research procedures received IACUC approval (Smithsonian Tropical Research Institute, assurance number 2012.0601.2015; University of California, Davis, assurance number 20442). We thank Ariana Strandburg-Peshkin, Alison Ashbury, and Niels Rattenborg for helpful comments on previous drafts of the manuscript. We thank M Wikelski, E Bermingham, D Rubenstein, M Kinnaird, D Carlino, Y Dellenback, S Dyer and B Tripard for logistical support, as well as R Kays, S Murray, M Mutinda, R Lessnau, S Alavi, J Nairobi, F Kuemmeth, and W Heidrich for field assistance. We are grateful to Shauhin Alavi for his statistical advice throughout the analysis, and to Vilson Karaj for carrying out behavioral observations for the validation study. We thank Yuuki Watanabe and an anonymous reviewer for providing productive comments and suggestions on the submitted manuscript that greatly improved the quality of the manuscript. This research received funding from the Max Planck Institute for Ornithology, the Smithsonian Tropical Research Institute, and University of California, Davis. RH and MCC were supported by the National Science Foundation (IIS 1514174 and IOS 1250895). MCC received additional support from a Packard Foundation Fellowship (2016-65130) and the Alexander von Humboldt Foundation in the framework of the Alexander von Humboldt Professorship endowed by the Federal Ministry of Education and Research. JCL was supported by a National Science Foundation Graduate Research Fellowship, the National Science Foundation Graduate Research Internship Program, a Dean's Distinguished Graduate Fellowship from the University of California, Davis, a Richard G Coss Wildlife Research Fellowship, and a scholarship from the Achievement Rewards for College Scientists (ARCS) Foundation. Further support for the project was also provided by the Center for the Advanced Study of Collective Behavior at the University of Konstanz, Deutsche Forschungsgemeinschaft Centre of Excellence 2117 (ID: 422037984).

## Additional information

### Funding

| Funder | Grant reference number | Author |
| --- | --- | --- |
| Max Planck Institute for Ornithology | | Margaret C Crofoot |
| Smithsonian Tropical Research Institute | | Margaret C Crofoot |
| University of California, Davis | | Margaret C Crofoot |
| Packard Foundation | 2016-65130 | Margaret C Crofoot |
| National Science Foundation | IIS 1514174 | Roi Harel<br>Margaret C Crofoot |
| National Science Foundation | IOS 1250895 | Margaret C Crofoot |

| Funder | Grant reference number | Author |
| --- | --- | --- |
| Alexander von Humboldt-Stiftung | Alexander von Humboldt Professorship endowed by the Federal Ministry of Education and Research | Margaret C Crofoot |
| National Science Foundation | Graduate Research Internship Program | J Carter Loftus |
| National Science Foundation | Graduate Research Fellowship | J Carter Loftus |
| Richard G. Coss Wildlife Research Fellowship | | J Carter Loftus |
| Center for the Advanced Study of Collective Behavior at the University of Konstanz | Deutsche Forschungsgemeinschaft Centre of Excellence 2117 (ID: 422037984) | J Carter Loftus Roi Harel Chase L Núñez Margaret C Crofoot |
| Achievement Rewards for College Scientists Foundation | | J Carter Loftus |

The funders had no role in study design, data collection and interpretation, or the decision to submit the work for publication.

## Author contributions

J Carter Loftus, Conceptualization, Formal analysis, Investigation, Methodology, Validation, Visualization, Writing – original draft, Writing – review and editing; Roi Harel, Conceptualization, Formal analysis, Investigation, Methodology, Supervision, Validation, Visualization, Writing – review and editing; Chase L Núñez, Investigation, Methodology, Supervision, Validation, Visualization, Writing – review and editing; Margaret C Crofoot, Conceptualization, Data curation, Funding acquisition, Investigation, Methodology, Project administration, Resources, Supervision, Validation, Writing – review and editing

## Author ORCIDs

J Carter Loftus (iD) http://orcid.org/0000-0002-0723-9159
Roi Harel (iD) http://orcid.org/0000-0002-9733-8643
Chase L Núñez (iD) http://orcid.org/0000-0002-3273-6966
Margaret C Crofoot (iD) http://orcid.org/0000-0002-0056-7950

## Ethics

All procedures were subject to ethical review and were carried out in accordance with the approved guidelines set out by the National Commission for Science, Technology and Innovation of the Republic of Kenya (NCTS/RCD/12B/012/26B, NACOSTI/P/15/5727/4608, NACOSTI/P/19/55517/24299, NACOSTI/P/20/1407). Baboon collaring and tracking was approved by the Smithsonian Tropical Research Institute (IACUC assurance number 2012.0601.2015), the University of California, Davis (IACUC assurance number 20442), and the Kenyan Wildlife Service (KWS/BRM/5001).

## Decision letter and Author response

Decision letter https://doi.org/10.7554/eLife.73695.sa1
Author response https://doi.org/10.7554/eLife.73695.sa2

# Additional files

## Supplementary files

• Supplementary file 1. Study metadata, detailed outputs from statistical analyses, and results of validation study. (a) Individual metadata. Table depicts the sex, age, weight, capture date, as well as data collection start and end dates for each study individual. F, female; M, male; A, adult; SA, subadult; J, juvenile; ACC, accelerometry. (b) Pearson correlation coefficient between the metrics of sleep extracted from the accelerometry data. Total sleep time is correlated with all sleep metrics, and sleep fragmentation is correlated with sleep efficiency. (c) Model output table of model of total

sleep time (for the first 20 days) with all numerical variables standardized. (d) Model output table of model of total sleep time (for the first 20 days) with no standardization of variables. (e) Model output table of model of total sleep time (for the first 20 days) with all numerical variables standardized (daytime vectorial dynamic body acceleration [VeDBA] included instead of travel distance). (f) Model output table of model of time spent napping during the day (for the first 20 days) with all numerical variables standardized. (g) Model output table of model of time spent napping during the day (for the first 20 days) without standardization of the variables. (h) Model output table of model of total sleep time using data from entire study duration (including after the leopard attack) with all variables standardized. (i) Model output table of model of total sleep time using data from entire study duration (including after the leopard attack) without standardization of variables. (j) Model output table of model of sleep fragmentation (for the first 20 days) with all numerical variables standardized. (k) Model output table of model of sleep fragmentation (for the first 20 days) with no standardization of variables. (l) Model output table of model of sleep fragmentation (for the first 20 days) with all numerical variables standardized (daytime VeDBA included instead of travel distance). (m) Model output table of model of sleep fragmentation using data from entire study duration (including after the presumed leopard attack) with all variables standardized. (n) Model output table of model of sleep fragmentation using data from entire study duration (including after the presumed leopard attack) without standardization of variables. (o) Model output table of model of synchronization (i.e., the proportion of minutes during a night that both dyad members exhibit the same behavior, either sleep or wakefulness) with response variable standardized. (p) Model output table of model of synchronization (i.e., the proportion of minutes during a night that both dyad members exhibit the same behavior, either sleep or wakefulness) without standardization of the response variable. (q) Model output table of model of an individual being awake in a given epoch. The previous night relative total sleep time and previous night relative sleep efficiency variables are standardized. Data that was modeled here was a subset of the full dataset, in which the focal baboon was not awake in the previous epoch, and the current epoch occurred between 21:00 and 05:00 and prior to the 15th day of the study, when several of the baboons' collars ceased collecting data. (r) Confusion matrix reporting the results of the validation study. Table entries represent the number of minute epochs classified according to the accelerometer-based technique and direct behavioral observation.

- Transparent reporting form

## Data availability
GPS and accelerometry data generated during this study are published and available in the Movebank repository (https://doi.org/10.5441/001/1.3q2131q5). Drone imagery is publicly available for download from Dryad (https://doi.org/10.5061/dryad.6h5b7). Accelerometry data and behavioral scoring data from the 2019 sleep validation study is also publicly available for download from Dryad (https://doi.org/10.5061/dryad.p5hqbzkqf). All code used to produce the final results from the raw data is on GitHub (https://github.com/CarterLoftus/baboon_sleep/) and archived on Zenodo (https://doi.org/10.5281/zenodo.5905742).

The following dataset was generated:

| Author(s) | Year | Dataset title | Dataset URL | Database and Identifier |
|---|---|---|---|---|
| Loftus JC, Harel R, Nuñez CL, Crofoot M | 2021 | Data from: Ecological and social pressures interfere with homeostatic sleep regulation in the wild | https://doi.org/10.5061/dryad.p5hqbzkqf | Dryad Digital Repository, 10.5061/dryad.p5hqbzkqf |

The following previously published datasets were used:

| Author(s) | Year | Dataset title | Dataset URL | Database and Identifier |
|---|---|---|---|---|
| Strandburg-Peshkin A, Farine D, Crofoot M, Couzin I | 2020 | Data from: Habitat and social factors shape individual decisions and emergent group structure during baboon collective movement | https://doi.org/10.5061/dryad.6h5b7 | Dryad Digital Repository, 10.5061/dryad.6h5b7 |
| Crofoot MC, Kays RW, Wikelski M | 2021 | Collective movement in wild baboons | https://doi.org/10.5441/001/1.3q2131q5 | Movebank Data Repository, 10.5441/001/1.3q2131q5 |

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
