## [Editor Report]

By recording sleep and movements in a group of baboons, this study reveals ecological and social drivers of sleep patterns in the wild. Using accelerometry, rather than methods that map brain activity, enables the investigation of animal sleep under natural conditions across a wide range of taxa.

---

## [Decision Letter]

**Decision letter after peer review:**

Thank you for submitting your article "Ecological and social pressures interfere with homeostatic sleep regulation in the wild" for consideration by *eLife*.

Your article has been reviewed by two peer reviewers, including Yuuki Y. Watanabe as Reviewing Editor and Reviewer #1, and the evaluation has been overseen by Christian Rutz as the Senior Editor.

The reviewers have discussed their reviews with one another, and the Reviewing Editor has drafted this decision letter to help you prepare a revised submission.

Essential revisions:

We have obtained two reviews, including one by the Reviewing Editor. Both reviewers agree that this study highlights exciting opportunities for studying sleep of animals under natural conditions, and thus, will be of general interest to behavioural ecologists. That said, we identified several issues that need to be addressed before we can offer publication in *eLife*.

(1) Baboons may be able to compensate shorter sleep with deeper sleep. Please explore the acceleration data in more detail to examine if sleep intensity can be inferred. If this is impossible, and there is no information on sleep intensity, the findings need to be interpreted more cautiously.

(2) Please provide more information on the leopard attack you observed.

(3) Please note that *eLife* has recently adopted the STRANGE framework, to help improve reporting standards and reproducibility in animal behaviour research. In your revision, please consider scope for sampling biases (selection of group, selection of group members for tagging, recovery of data from tagged subjects etc.) and potential limitations to the generalisability of your findings:

*Reviewer #1 (Recommendations for the authors):*

L168-170 These sentences are an interpretation rather than result. Results should be reported in more objective manners (e.g., sleeping time during the prior night did not affect…). This comment also applied to other parts in the Results.

L191 How did you know that a leopard attacked?

L224 This sentence seems to be an overstatement. Even though some baboons have to decrease sleep time due to ecological/social demands, they may still maintain homeostasis.

L261 Lyamin et al., (2018) did not study fur seals' sleep during oceanic migrations.

*Reviewer #2 (Recommendations for the authors):*

My main concern is that sleep intensity was not measured, and therefore the authors do not have a full accounting of the sleep obtained by the baboons. This is a common limitation shared by all accelerometry-based studies of sleep. One way to partially address whether sleep intensity varied in response to the ecological parameters measured would be to examine the accelerometry data in finer detail. For example, it is possible that baboons that are sleeping deeper have fewer small awakenings. Consequently, activity detected below the awake threshold used in the current paper might vary in response to ecological factors; i.e., baboons that are sleeping deeper would be expected to have less sub-threshold activity.

The fact that a leopard attack occurred during the study is very interesting, as very little is known about how animals adjust their sleep in response to predation attempts. However, to fully interpret the baboons' response, it would be helpful to have more information about the attack. How was it determined that an attack occurred? When did it happen? Did it involve the leopard entering a sleeping tree? If so, did the baboons in that tree sleep differently from those in nearby trees? Did the baboons know about the attack when it happened, or did they just discover that a leopard had been nearby when they left the tree in the morning? The authors should also consider the possibility that the baboons spent more time sitting motionless and awake following the attack, a behavior that would register as sleep on the accelerometer.

Regarding the validation study, it would be helpful if the authors included images of the baboons exhibiting the various behaviors scored from the images.

Comments regarding references:

Line 92: When referring to animals that can engage in sleep while doing other things by sleeping unihemispherically, in addition to Rattenborg et al., 2016, the authors should also reference Lyamin et al., Current Biology, 2018 for fur seals, Lyamin et al., Neurosci. Biobehav. Rev., 2008 for cetaceans, and Rattenborg et al., Nature 1999 for ducks.

Line 110: Regarding sleep disruption being caused by nearby group members, the authors should take a look at these articles which describe related behaviors in gulls sleeping in a flock:

https://royalsocietypublishing.org/doi/10.1098/rsbl.2008.0490

https://psycnet.apa.org/record/2011-05106-007

Line 261: It is important to note that the fur seals in Lyamin et al., 2018 were only studied in captivity.

---

## [Author Response]

Essential revisions:We have obtained two reviews, including one by the Reviewing Editor. Both reviewers agree that this study highlights exciting opportunities for studying sleep of animals under natural conditions, and thus, will be of general interest to behavioural ecologists. That said, we identified several issues that need to be addressed before we can offer publication in eLife.(1) Baboons may be able to compensate shorter sleep with deeper sleep. Please explore the acceleration data in more detail to examine if sleep intensity can be inferred. If this is impossible, and there is no information on sleep intensity, the findings need to be interpreted more cautiously.

We agree with the reviewers that addressing sleep intensity/depth is necessary to support the conclusions of our manuscript. The lack of established and open-sourced methods to measure sleep depth from accelerometry limits the scope of investigations into sleep that rely exclusively on this method. Fortunately (and thanks to this push from our reviewers), we were able to leverage our finding that baboons wake in response to the waking of nearby group-mates to assess arousal thresholds during sleep, which as Reviewer 2 remarks, is the most direct measure of sleep intensity. We tested the hypothesis that baboons maintain sleep homeostasis by sleeping deeper (even if not longer) following periods of poor sleep. To test this hypothesis, we modelled the probability of a baboon waking in response to the nighttime activity of a group-mate in his/her sleep tree as a function of the duration and fragmentation of sleep on the previous night, expecting that if baboons sleep deeper following nights of poor sleep, they would be less likely to be awoken by the nocturnal waking of their neighbors (i.e. they would have a higher arousal threshold) following nights of short duration and/or highly fragmented sleep. We found that although baboons are indeed much more likely to awaken when a nearby group-mate was active in the previous minute epoch, their responsiveness, and thus arousal threshold, was not modulated by the previous night’s total sleep time or sleep fragmentation. Thus, they did not sleep deeper following nights of poor sleep. We have detailed our statistical approach in the Materials and methods, reported these findings in the Results sections, and addressed the implications in the Discussion. We have also made an adjustment to the Abstract based on this analysis.

To further explore sleep intensity in our data, we drew from the Reviewer 2’s suggestion that baboons that are sleeping deeper may have fewer small awakenings and evidence that transitions from deeper to lighter stages of sleep are often accompanied by large body movements (Muzet et al., 1972, 2016; Muzet and Schieber, 1971), and we analyzed periods of movement that occurred within the sleep period. A greater number of distinct periods of night-time movement would indicate a greater number of sleep stage transitions, and likely fewer epochs of deep sleep. We added a full analysis of sleep fragmentation (measured as the number of distinct bouts greater than 2 minutes during the sleep period, divided by the total sleep time, following Samson and Nunn, 2015), mirroring that of total sleep time, to the

Results section, and made relevant adjustments to the Discussion, Materials and methods, and Supplemental Information sections. We also included the fragmentation of the previous night’s sleep as a predictor in our models, and updated the Results and Supplemental Information accordingly.

Although sleep fragmentation can indeed correlate with sleep depth (Bastuji and García-Larrea, 1999), our literature search did not reveal many analyses of the link between sleep fragmentation and depth (which we found surprising, given that many studies of sleep have the data to easily test for a correlation between these variables). Thus, this additional analysis is perhaps best viewed as complementary to our direct analysis of arousal thresholds (via probability of waking in response to the night-time activity of a nearby group-mate), rather than as a stand-alone measure of sleep depth. Accordingly, in the manuscript, we did not draw any conclusions about sleep depth that arose from the sleep fragmentation analysis alone.

Reviewer 2 also offered other helpful suggestions as to how we might go about inspecting the accelerometry data in finer detail to measure sleep depth – specifically, by inspecting the activity that occurred below the threshold to be considered sleep. We analyzed the log VeDBA, the measure of activity that we employed in this study, within the epochs that were classified as sleep by the accelerometer-based sleep classification algorithm, hoping to find multi-modally distributed data that indicated unique activity states within the epochs classified as sleep. Unfortunately, we found a highly normal distribution (Author response image 1) that likely represents slight noise around a single mean log VeDBA value indicative of immobility. We were, therefore, not confident about our ability to extract meaningful information about the depth of sleep from this measure of activity below the sleep threshold.

**Author response image 1. sa2fig1:** 

Winnebeck and colleagues (2018) show that a non-linear transformation of activity data to an account of inactivity – termed the Locomotor Inactivity During Sleep (LIDS) – can enable the extraction of sleep depth from accelerometry data produced by wrist-borne devices in humans. However, the nonlinear transformation requires count data from an actigraph, not raw accelerometry. Actigraph count data are the product of onboard processing of raw data via a proprietary algorithm, which complicates replication of these methods. If we approximated count data from our raw accelerometry (e.g. by following Brønd et al., 2017 or Brondeel et al., 2021), we would still be unable to validate the relationship between LIDS and arousal threshold or its correlated sleep stages in baboons. New technologies and algorithms (e.g. the Zulu watch (Devine et al., 2021) and others reviewed by Imtiaz, 2021) are also able to extract information about the depth of sleep solely from motion data produced by a wrist-borne sensor.

Unfortunately, the proprietary nature of both the hardware and software employed by these devices limits their usefulness in our study.

(2) Please provide more information on the leopard attack you observed.

We agree that more information about the leopard attack aids in the interpretation of the baboons’ response to the attack, and we thank the reviewers for their suggestion. We added further details about the attack in both the Results (lines 217 – 223 [208 – 213]) and the Materials and methods (lines 466 – 467 [441 – 451]). In these additions, we describe that we concluded that an attack had occurred after hearing snarls and growls followed by sustained and intense baboon screaming and alarm calling. After comparing the vocalizations to recordings of the vocalizations of the large cats that are at the sleep site, and with previous literature suggesting that leopards are the only predator that readily attack baboons in their sleep site (Busse, 1980; Cheney et al., 2004), we concluded that a leopard had launched the attack. The attack occurred well after sunset (sunset: 18:41; attack: 20:55), and as such, we were unable to visually observe the predator or the baboons during the attack. Because we could not visually observe the attack, we unfortunately were unable to analyze in more detail the impacts of a predation attempt on sleep patterns (e.g. did baboons that were targeted by the predator sleep differently from other group members?). The surprising lack of movement in response to the attack, as indicated by the accelerometry data, also prevented us from using the movement data to retroactively infer the targets of the attack.

Reviewer 2 also brought our attention to the possibility that baboons may have sat motionless, but awake, following the attack, which would be classified as sleep by the accelerometer-based sleep classifier. We thank the reviewer for this contribution, and we have added a consideration of this possibility in the Discussion (lines 304 – 307 [288 – 291]).

(3) Please note that eLife has recently adopted the STRANGE framework, to help improve reporting standards and reproducibility in animal behaviour research. In your revision, please consider scope for sampling biases (selection of group, selection of group members for tagging, recovery of data from tagged subjects etc.) and potential limitations to the generalisability of your findings:

To adhere more strictly to *eLife* policy, we have expanded Table S1 (now Supplementary file 1a) to include the body mass, age class, sex, and date of trapping/collaring of each study individuals.

We have also added a section to the Materials and methods (lines 713 – 775 [684 – 746]) to thoroughly declare and discuss all potential biases that may influence our results or limit the generalizability of our findings. We feel that, broadly, our findings are generalizable and that the influence of sampling biases on our conclusions is minimal. However, our selection of study subjects based on size presents a very real bias against including data from infants and young juveniles in our sample. This bias may have critically impacted our finding that older individuals slept more than younger individuals. In our revised version of the manuscript, we draw attention to this bias, and its potential impact on our results in the Discussion section (lines 401 – 406 [378 – 383]).

Reviewer #1 (Recommendations for the authors):L168-170 These sentences are an interpretation rather than result. Results should be reported in more objective manners (e.g., sleeping time during the prior night did not affect…). This comment also applied to other parts in the Results.

We edited this sentence accordingly. We also edited several other sentences of the Results to reflect more objectivity and remove interpretation.

L191 How did you know that a leopard attacked?

See response to Essential Revision 2.

L224 This sentence seems to be an overstatement. Even though some baboons have to decrease sleep time due to ecological/social demands, they may still maintain homeostasis.

The reviewer makes a good point that the baboons likely maintain sleep homeostasis on a larger scale, even if they do often prioritize ecological and social demands over fulfilling sleep requirements. We have adjusted the wording of this sentence accordingly.

L261 Lyamin et al., (2018) did not study fur seals' sleep during oceanic migrations.

We thank the reviewer for catching our mistake here in wording this sentence such that it suggests that the study performed by Lyamin and colleagues (2018) was carried out in the wild. We have removed this example and its citation from this part of the manuscript.

Reviewer #2 (Recommendations for the authors):My main concern is that sleep intensity was not measured, and therefore the authors do not have a full accounting of the sleep obtained by the baboons. This is a common limitation shared by all accelerometry-based studies of sleep. One way to partially address whether sleep intensity varied in response to the ecological parameters measured would be to examine the accelerometry data in finer detail. For example, it is possible that baboons that are sleeping deeper have fewer small awakenings. Consequently, activity detected below the awake threshold used in the current paper might vary in response to ecological factors; i.e., baboons that are sleeping deeper would be expected to have less sub-threshold activity.

See response to Essential Revision 1.

The fact that a leopard attack occurred during the study is very interesting, as very little is known about how animals adjust their sleep in response to predation attempts. However, to fully interpret the baboons' response, it would be helpful to have more information about the attack. How was it determined that an attack occurred? When did it happen? Did it involve the leopard entering a sleeping tree? If so, did the baboons in that tree sleep differently from those in nearby trees? Did the baboons know about the attack when it happened, or did they just discover that a leopard had been nearby when they left the tree in the morning? The authors should also consider the possibility that the baboons spent more time sitting motionless and awake following the attack, a behavior that would register as sleep on the accelerometer.

See response to Essential Revision 2.

Regarding the validation study, it would be helpful if the authors included images of the baboons exhibiting the various behaviors scored from the images.

We thank the reviewer for this helpful suggestion, and we have added the suggested images to the Supplemental Information (Figure 5—figure supplement 1). We would be happy to move this figure to the Materials and methods section if the reviewers would find that more appropriate.

Comments regarding references:Line 92: When referring to animals that can engage in sleep while doing other things by sleeping unihemispherically, in addition to Rattenborg et al., 2016, the authors should also reference Lyamin et al., Current Biology, 2018 for fur seals, Lyamin et al., Neurosci. Biobehav. Rev., 2008 for cetaceans, and Rattenborg et al., Nature 1999 for ducks.

We thank the reviewer for the additional references, and have added them to the manuscript as suggested.

Line 110: Regarding sleep disruption being caused by nearby group members, the authors should take a look at these articles which describe related behaviors in gulls sleeping in a flock:https://royalsocietypublishing.org/doi/10.1098/rsbl.2008.0490https://psycnet.apa.org/record/2011-05106-007

We thank the reviewer for bringing these relevant studies to our attention. We have consulted them, and we reference them in this revised version of the manuscript.

Line 261: It is important to note that the fur seals in Lyamin et al., 2018 were only studied in captivity.

We thank the reviewer for catching our mistake here in wording this sentence such that it suggests that the study performed by Lyamin and colleagues (2018) was carried out in the wild. We have decided that extending the length of the manuscript to add the necessary clarification here would not be worthwhile, seeing that the example is not essential to the paragraph. We have therefore removed this example and its citation from this paragraph.

References

Bastuji, H., & García-Larrea, L. (1999). Sleep/wake abnormalities in patients with periodic leg movements during sleep: Factor analysis on data from 24-h ambulatory polygraphy. Journal of Sleep Research, 8(3), 217–223. https://doi.org/10.1046/j.1365-2869.1999.00157.x

Brønd, J. C., Andersen, L. B., & Arvidsson, D. (2017). Generating actiGraph counts from raw acceleration recorded by an alternative monitor. 2351-2360. https://doi.org/10.1249/MSS.0000000000001344

Brondeel, R., Kestens, Y., Anaraki, J. R., Stanley, K., Thierry, B., & Fuller, D. (2021). Converting Raw Accelerometer Data to Activity Counts Using Open-Source Code: Implementing a MATLAB Code in Python and R, and Comparing the Results to ActiLife. Journal for the Measurement of Physical Behaviour, 1(aop), 1–7.

Busse, C. (1980). Leopard and lion predation upon chacma baboons living in the Moremi Wildlife Reserve. Botswana Notes and Records, 15–21.

Cheney, D. L., Seyfarth, R. M., Fischer, J., Beehner, J., Bergman, T., Johnson, S. E., Kitchen, D. M., Palombit, R. A., Rendall, D., & Silk, J. B. (2004). Factors affecting reproduction and mortality among baboons in the Okavango Delta, Botswana. International Journal of Primatology, 25(2), 401–428.

Devine, J. K., Chinoy, E. D., Markwald, R. R., Schwartz, L. P., & Hursh, S. R. (2021). Validation of Zulu Watch against Polysomnography and Actigraphy for On-Wrist Sleep-Wake Determination and Sleep-Depth Estimation. Sensors, 21(1), 76. https://doi.org/10.3390/s21010076

Imtiaz, S. A. (2021). A Systematic Review of Sensing Technologies for Wearable Sleep Staging. Sensors, 21(5), 1562. https://doi.org/10.3390/s21051562

Lyamin, O. I., Kosenko, P. O., Korneva, S. M., Vyssotski, A. L., Mukhametov, L. M., & Siegel, J. M. (2018). Fur Seals Suppress REM Sleep for Very Long Periods without Subsequent Rebound. Current Biology, 28(12), 2000-2005.e2. https://doi.org/10.1016/j.cub.2018.05.022

Muzet, A., Naitoh, P., Townsend, R., & Johnson, L. (1972). Body movements during sleep as a predictor of stage change. Psychon Sci, 29, 7–10.

Muzet, A., & Schieber, J. P. (1971). Etude de la covariation de variables électroencéphalographiques, motrices et cardiaque au cours du sommeil nocturne chez l’Homme. Arch Sci Physiol, 25, 467–516.

Muzet, A., Werner, S., Fuchs, G., Roth, T., Saoud, J. B., Viola, A. U., Schaffhauser, J.-Y., & Luthringer, R. (2016). Assessing sleep architecture and continuity measures through the analysis of heart rate and wrist movement recordings in healthy subjects: Comparison with results based on polysomnography. Sleep Medicine, 21, 47–56. https://doi.org/10.1016/j.sleep.2016.01.015

Samson, D. R., & Nunn, C. L. (2015). Sleep intensity and the evolution of human cognition. Evolutionary Anthropology: Issues, News, and Reviews, 24(6), 225–237. https://doi.org/10.1002/evan.21464

Winnebeck, E. C., Fischer, D., Leise, T., & Roenneberg, T. (2018). Dynamics and Ultradian Structure of Human Sleep in Real Life. Current Biology, 28(1), 49-59.e5. https://doi.org/10.1016/j.cub.2017.11.063